# Phosphodiesterase 4D acts downstream of Neuropilin to control Hedgehog signal transduction and the growth of medulloblastoma

Xuecai Ge[1], Ljiljana Milenkovic[1], Kaye Suyama[1], Tom Hartl[1], Teresa Purzner[1], Amy Winans[2], Tobias Meyer[2], Matthew P Scott[1]*

[1]Department of Developmental Biology, Department of Genetics, Department of Bioengineering, Stanford University School of Medicine, Stanford, United States; [2]Department of Chemical and Systems Biology, Stanford University School of Medicine, Stanford, United States

*For correspondence: mscott@stanford.edu.

Competing interests: The authors declare that no competing interests exist.

**Abstract** Alterations in Hedgehog (Hh) signaling lead to birth defects and cancers including medulloblastoma, the most common pediatric brain tumor. Although inhibitors targeting the membrane protein Smoothened suppress Hh signaling, acquired drug resistance and tumor relapse call for additional therapeutic targets. Here we show that phosphodiesterase 4D (PDE4D) acts downstream of Neuropilins to control Hh transduction and medulloblastoma growth. PDE4D interacts directly with Neuropilins, positive regulators of Hh pathway. The Neuropilin ligand Semaphorin3 enhances this interaction, promoting PDE4D translocation to the plasma membrane and cAMP degradation. The consequent inhibition of protein kinase A (PKA) enhances Hh transduction. In the developing cerebellum, genetic removal of Neuropilins reduces Hh signaling activity and suppresses proliferation of granule neuron precursors. In mouse medulloblastoma allografts, PDE4D inhibitors suppress Hh transduction and inhibit tumor growth. Our findings reveal a new regulatory mechanism of Hh transduction, and highlight PDE4D as a promising target to treat Hh-related tumors.

## Introduction

The Hedgehog (Hh) pathway is essential in the morphogenesis and patterning of many tissues (*Briscoe and Therond, 2013*). Hh signaling is transduced through a series of negative regulatory interactions. Binding of a Hh ligand such as Sonic hedgehog (Shh) to its receptor Patched (Ptch) releases Ptch inhibition of Smoothened (Smo). Activated Smo then triggers a signaling cascade that eventually activates Gli transcription factors, which enable Hh target gene transcription (*Briscoe and Therond, 2013*). Abnormalities in Hh signaling are responsible for certain birth defects and contribute to a wide range of tumors, including medulloblastoma (MB), the most common malignant pediatric brain tumor (*Taylor et al., 2012*). For example, the loss of the negative regulator Ptch, causes Smo to be inappropriately active, and this leads to skin cancer and MB (*Johnson et al., 1996*; *Stone et al., 1996*).

Current treatment for MB, surgical removal followed by chemotherapy and radiotherapy, cures about 2-thirds of patients, but often leaves survivors suffering from devastating neurocognitive consequences (*Fouladi et al., 2005*). The Smo inhibitor, vismodegib, is the only FDA-approved drug that targets the Hh pathway. However, the effect of vismodegib tends to be transient, and irreversible drug resistance and tumor relapse often ensue (*Rudin et al., 2009*; *Yauch et al., 2009*). Genomic sequencing results showed that the resistance is due to mutations in Smo that reduce its binding

**eLife digest** A communication system in cells called the Hedgehog signaling pathway plays an essential role in the formation of tissues and organs in animal embryos. The activity of the pathway is carefully controlled during development and if Hedgehog signaling is disrupted it can lead to developmental defects and particular types of cancer. Some of these cancers can be treated with a drug called vismodegib, which targets a particular molecule in the Hedgehog signaling pathway. However, tumor cells can become resistant to this drug, so researchers are hoping to find new therapies that target other aspects of the signaling pathway.

Hedgehog signaling promotes the division of brain cells called granule neuron precursor cells (or GNP cells for short). If the signaling pathway is over-active it can trigger the GNP cells to divide more than they should. This can lead to medulloblastoma, which is the most common type of brain tumor that affects children. Proteins called Neuropilins—which bind to molecules known as Semaphorins—promote Hedgehog signaling and the formation of medulloblastoma, but it was not clear how this works.

Here Ge et al. studied the role of Neuropilin in cultured cells and in the cerebellum of mice. The experiments show that Semaphorin 3 promotes the accumulation of an enzyme called PDE4D at the cell membrane. PDE4D interacts with Neuropilin and blocks the activity of another enzyme that normally inhibits Hedgehog signaling. In mice that lack Neuropilin and Semophorin 3, the GNP cells are less able to divide, which leads to abnormal development of the cerebellum.

Further experiments show that drugs that target PDE4D inhibit both the Hedgehog pathway and the growth of tumors that are resistant to vismodegib treatment. Ge et al.'s findings uncover a new way in which Hedgehog signaling is regulated and highlight a potential new strategy for treating medulloblastoma and other similar tumors. Current PDE4D inhibitors are associated with severe side effects, so the next challenge is to develop new drugs that have fewer side effects.

---

affinity to vismodegib, or cause Smo to be constitutively active in tumor cells. Thus novel drug targets downstream of Smo are critically needed. We previously discovered that Neuropilins (Nrp1&2) positively regulate Hh transduction downstream of Smo (*Hillman et al., 2011*). Nrps are co-receptors for Semaphorin 3 (Sema3) in axon guidance (*Gu et al., 2002*; *Appleton et al., 2007*), and for VEGF that controls cardiovascular development and angiogenesis (*Pellet-Many et al., 2008*). Nrps are also widely involved in many types of cancers. In particular, they are implicated in the growth and spread of MB (*Hayden Gephart et al., 2013*; *Snuderl et al., 2013*). However, how Nrps regulate Hh transduction at the molecular level remains unknown, which limits our understanding of developmental signal integration and hampers the development of Nrp-related therapeutics for Hh-related tumors.

MB results from the over-proliferation of granule neuron precursors (GNPs) in the developing cerebellum. GNP proliferation during normal development is stimulated by Hh signaling. Shh released by Purkinje neurons acts as a mitogen for GNPs (*Dahmane and Ruiz i Altaba, 1999*; *Wallace, 1999*; *Wechsler-Reya and Scott, 1999*). Activating mutations in the Hh pathway are responsible for about one-quarter of MB cases. *Nrp1* and *Nrp2* are highly expressed in developing cerebellum and in human MBs (*Snuderl et al., 2013*). One way that Nrps can affect tumor growth is by facilitating VEGF-driven vascularization events that prevent tumor suffocation. However, several lines of evidence suggest that Nrps promote tumor growth through mechanisms in addition to VEGF-mediated angiogenesis. Abolishing Nrp function in tumor cells using function-blocking antibodies suffices to suppress MB growth and metastasis (*Snuderl et al., 2013*). Selective inhibition of Nrp function by RNAi specifically in MB tumor cells, that is not in vasculature, blocks growth of MB allografts (*Hayden Gephart et al., 2013*). However, despite extensive research on each pathway, it remains unclear whether and how Nrps interact with Hh pathways to control GNP proliferation and MB formation.

Here we describe a molecular mechanism that integrates Sema3/Nrp signaling with Hh transduction. We found that Sema3, a member of a secreted ligand family, enhances Hh signaling. This is achieved by activation of Phosphodiesterase 4D (PDE4D), which reduces intracellular cAMP levels through hydrolysis. The subsequent inhibition of Protein Kinase A (PKA) activity promotes Hh transduction. We demonstrate that this molecular interplay operates in the developing cerebellum.

Genetic removal of Sema3/Nrp signaling severely impairs GNP proliferation. Furthermore, inhibiting PDE4D suppresses the growth of Hh-related MB. These findings reveal a hitherto unknown transduction mechanism that links Sema3/Nrp signaling with Hh pathway, 2 major pathways in development and disease, and highlight PDE4D as a new therapeutic target for Hh-related tumors.

## Results

### The Nrp A1/A2 and cytoplasmic domains are required to enhance Hh signal transduction

Nrps are single transmembrane proteins with five extracellular domains and a short cytoplasmic tail. The binding of Sema3 and VEGF to Nrps is ascribed to the extracellular A1/A2 and B1/B2 domains, respectively (*Figure 1A*). Little is known about the function of the Nrp cytoplasmic domain, though it is evolutionarily conserved. To identify which domain of Nrps is required to promote Hh transduction, we silenced endogenous Nrps in NIH3T3 cells with lentivirus-mediated RNAi and over-expressed truncated Nrp1 to rescue the Hh signaling. We found that both extracellular and cytoplasmic domains are required to promote Hh signaling, since Nrp mutants without either domain were expressed stably but failed to rescue Hh transduction (*Figure 1B*). At the extracellular side, A1/A2 domains are required for Nrps to enhance Hh transduction, whereas B1/B2 domains are dispensable (*Figure 1B*). We then explored what Hh transduction events are mediated by the requisite domains.

### Sema3 enhances Hh signal transduction

The Nrp A1/A2 domains interact with Sema3 secreted protein family. We therefore tested the effect of Sema3 on Hh transduction. We treated NIH3T3 cells with Sema3A or 3F, two well-characterized ligands for Nrp1 and 2, respectively. An increasing concentration of Shh was used to induce Hh target gene (*Gli1*) expression. Both Sema3A and 3F dramatically enhanced Shh-induced transcription of *Gli1* (*Figure 1C*). The most prominent amplification of Shh response was seen when Sema3F was added to cells in the presence of a high concentration of Shh (2 µg/ml). Sema3A and 3F were not able to induce *Gli1* expression in the absence of Shh. Three other Sema3 isoforms, 3B, 3C and 3E, had similar effects (*Figure 1—figure supplement 1A*). Similar results were obtained when *Gli1* was induced by SAG, a small molecule that binds directly to Smo and triggers target gene expression (*Figure 1—figure supplement 1A*). Together, these results demonstrate that activities of proteins in the Sema3 family contribute to Hh transduction. Since a single isoform of the Sema3 family is able to promote Hh transduction, blocking all Sema3 isoforms may be necessary in order to fully compromise Hh signaling. We performed such experiments by silencing both Nrps and by employing antibodies that block the Sema3-Nrp interaction (*Figure 1F*, *Figure 1—figure supplement 1F,G*). Future studies using genetic approaches (e.g., genetic knockdown or knockout) to lower multiple Sema3 ligands will be required to further define the role of Sema3 proteins in enhancing Hh signaling. In our previous study, using NIH3T3 cells stably transfected with a Gli-responsive luciferase reporter (Shh-Light2 cells), we did not find a noticeable effect of Sema3A on Hh transduction (*Hillman et al., 2011*). In the current study, we used unmodified NIH3T3 cells and new batches of recombinant Sema3, and employed a more sensitive method (qPCR) to detect Hh signal activity. With this approach we detected robust Sema3-mediated enhancement of Hh signaling.

In contrast to the strong Sema effects, recombinant VEGF had no impact on Shh-induced Gli1 transcription (*Figure 1D*). To make sure that the inactivity of VEGF was not due to absence of VEGF receptor, a co-receptor with Nrps in angiogenesis, we confirmed expression of VEGF receptor (VEGFR1) in NIH3T3 cells (*Figure 1—figure supplement 1B*). Taken together, we found a hitherto unknown synergy between Sema3 and Hh transduction.

### The binding of Sema3-Nrp is required to enhance Hh signal transduction

In NIH3T3 cells, we detected the expression of Sema3F by Western blot (*Figure 1—figure supplement 1C*). We also detected the expression of other Sema3 isoforms by next generation RNA sequencing (data not shown). It is likely that these endogenous Sema3 contribute to the baseline level of Hh transduction. To block the binding of the endogenous Sema3 with Nrps, we used function-blocking antibodies that specifically blocked ligand-binding sites in Nrps. The antibody Nrp[panA] occupies Sema3's binding site in both Nrp1 and Nrp2; antibodies Nrp[1B] and Nrp[2B] occupy VEGF's

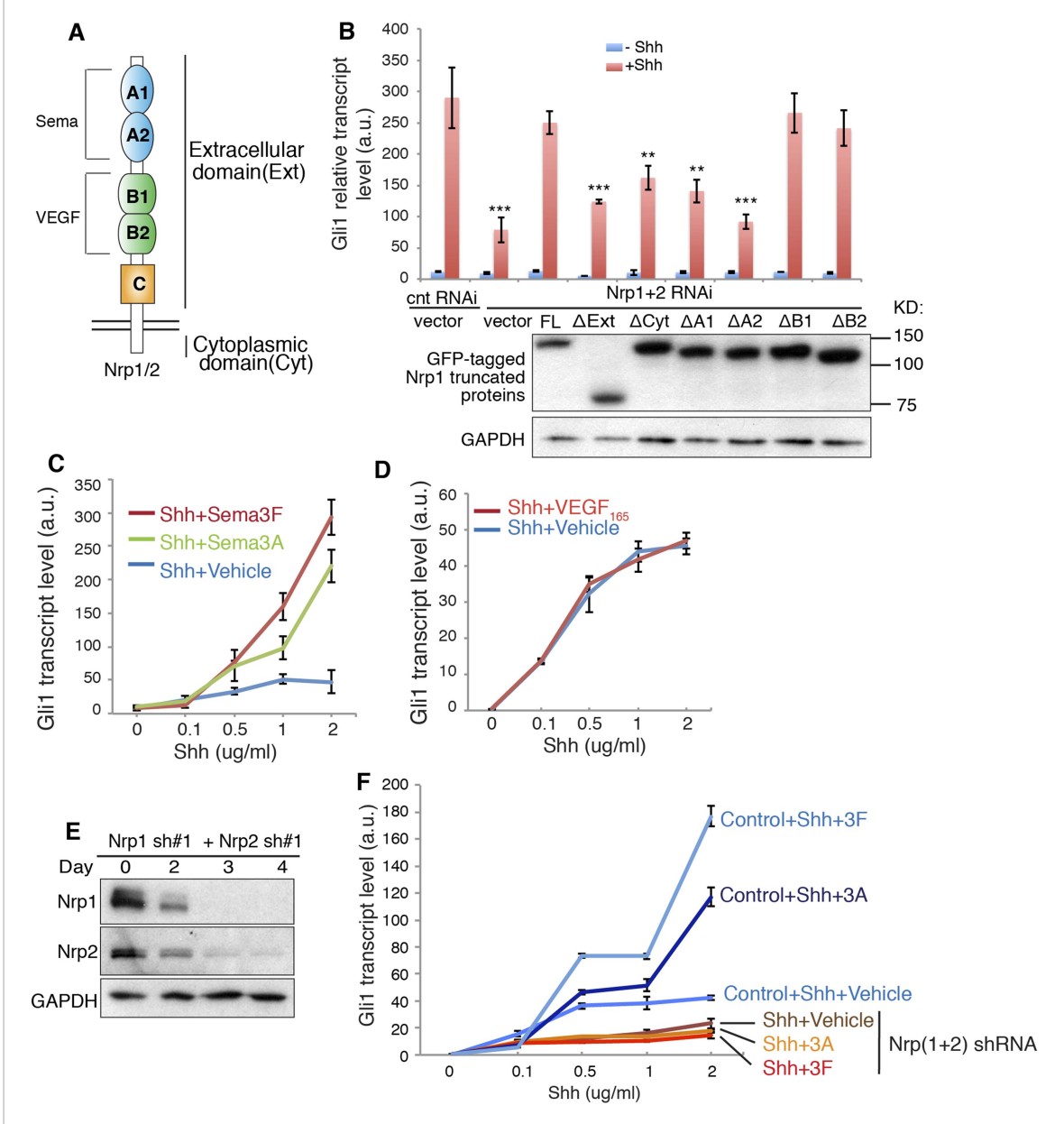

**Figure 1**. Signaling downstream of Sema3-Nrp enhances Hh transduction. (**A**) Schematic drawing of Nrps protein structure. (**B**) In NIH3T3 cells in which Nrp1&2 were silenced by RNAi, Hh signaling could be rescued by full-length (FL) Nrp1 construct, but not by Nrp1 constructs that lack the entire extracellular domain (ΔExt), cytoplasmic domain (ΔCyt), A1 (ΔA1), or A2 (ΔA2) domain. Western blot shows that truncated Nrps were expressed with expected molecular weights. (**C, D**) Hh signaling activity in NIH3T3 cells treated with increasing concentrations of recombinant Shh in conjunction with a constant concentration of Sema3A, Sema3F (3 μg/ml), or VEGF165 (100 ng/ml) for 24 hr (**E, F**) Western blot shows that lentivirus-mediated expression of shRNA against Nrp1 and Nrp2 abolished the expression of endogenous Nrps (**E**). On day 3, Hh signaling activity was evaluated after cells were treated with Shh in conjunction with Sema3A or Sema3F for 24 hr (**F**). In all experiments Gli1 transcript level was measured by qPCR to evaluate Hh signaling activity; a.u., arbitrary unit. All error bars represent SEM. Statistics: Student's t-Test. *p < 0.05.

The following figure supplement is available for figure 1:

**Figure supplement 1**. Sema3 signals through Nrps to enhance Hh transduction.

binding sites in Nrp1 and Nrp2, respectively (*Appleton et al., 2007*). These antibodies were previously used to block tumor growth by interfering with tumor vascularization (*Liang et al., 2007*; *Pan et al., 2007*). We treated NIH3T3 cells with each antibody together with Shh or SAG. As expected, Nrp[panA] significantly reduced Shh- and SAG-induced *Gli1* transcription, whereas Nrp[1B] and Nrp[2B] had no significant effect (*Figure 1—figure supplement 1D–F*). Thus blocking Sema3F from binding to Nrp prevented Sema3F from enhancing Hh transduction, while control antibodies did not affect Hh transduction.

We then silenced expression of both *Nrp* genes with shRNA-expressing lentiviruses (*Figure 1F*). As reported before (*Hillman et al., 2011*), Nrp loss reduced *Gli1* transcription to ~30% of the control level. Furthermore, Sema3A and 3F failed to enhance Shh-induced target gene transduction in the absence of Nrps (*Figure 1E–F*). To exclude possible off-target effects of RNAi, we used another pair of ShRNAs to silence *Nrp* expression, and got similar results (*Figure 1—figure supplement 1G,H*). These data suggest that Sema3 acts through receptor Nrps to promote Hh signal transduction.

## Nrp1 interacts with PDE4D, and Sema3 promotes this interaction

The short cytoplasmic domains of Nrps have had no clear role in transducing any signal. However, recently this domain was found to be required for the survival and growth of MB cells (*Snuderl et al., 2013*), and our data suggest that this domain is indispensable for robust Hh transduction (*Figure 1B*). To discover how the cytoplasmic domains of Nrps may operate in the Hh pathway, we performed yeast two-hybrid screening using the Nrp1 cytoplasmic domain as the 'bait'. Phosphodiesterase 4D (PDE4D) emerged multiple times from the screen. PDE4D belongs to the superfamily of PDE4, enzymes that degrade the second messenger cyclic AMP (cAMP). Multiple isoforms of PDE4D with distinct molecular weights are generated through alternative RNA splicing (*Richter et al., 2005*). We detected three main isoforms expressed in NIH3T3 cells: PDE4D2/6 (68KD), PDE4D1 (78KD) and PDE4D4 (110KD) (*Figure 2—figure supplement 1A*). We then used co-immunoprecipitation to determine whether the interaction between Nrp1 and PDE4D observed in yeast is indicative of an interaction in NIH3T3 cells. We found that Nrp1 interacted with PDE4D, in particular the isoform PDE4D2/6 (*Figure 2A*).

Since Nrp1 and Nrp2 play partially redundant roles in the regulation of Hh transduction (*Hillman et al., 2011*), we tested whether both Nrps interact with PDE4D. We overexpressed the tagged protein Nrp1/2-EGFP and PDE4D2-Flag, and found that both Nrps immunoprecipitated PDE4D2 (*Figure 2—figure supplement 2A,B*). To further determine the involvement of the cytoplasmic domain of Nrps in the interaction with PDE4D, we deleted this domain from both Nrps. The truncated Nrp proteins failed to immunoprecipitate PDE4D2. Thus, the interactions of PDE4D2 with Nrp1 and Nrp2 depends upon their cytoplasmic domains.

The interaction of the Nrp cytoplasmic domain with PDE4D could be related to the Sema3 signaling that affects Hh transduction. To test the connection, NIH3T3 cells were treated with Sema3F, and Nrp1 was immunoprecipitated. Sema3F treatment increased the amount of PDE4D2/6 that was immunoprecipitated by Nrp1 antibody in a time-dependent manner (*Figure 2B–C*). Thus binding of Sema3F to Nrp1 facilitates the interaction between Nrp1 and PDE4D.

## Sema3, through Nrps, increases PDE4D recruitment to the cell membrane and reduces intracellular cAMP level

Nrp1 is a transmembrane protein, primarily located in the plasma membrane. We hypothesized that through interacting with Nrp1, PDE4D is recruited to the plasma membrane. Since Sema3F promotes the Nrp-PDE4D interaction, the net effect would be that Sema3 promotes plasma membrane localization of PDE4D. We tested this hypothesis by subcellular fractionation, examining the intensity of PDE4D2/6 in the cytosolic and membrane fractions. 30 min after cells were treated with Sema3F, the amount of PDE4D2/6 in the membrane fraction increased significantly, while the amount in the cytosolic fraction decreased slightly (*Figure 2D–E*). Note that the time scale is in agreement with the co-immunoprecipitation; recruitment of PDE4 to the membrane coincides with the increased co-immunoprecipitation (*Figure 2B–C*). As a conclusion, Sema3 promotes recruitment of PDE4D to the cell membrane.

To test the effect of Shh on Sema3-mediated PDE4D translocation to the cell membrane, we treated cells with Shh in the presence or absence of Sema3 prior to cell fractionation. Shh by itself did

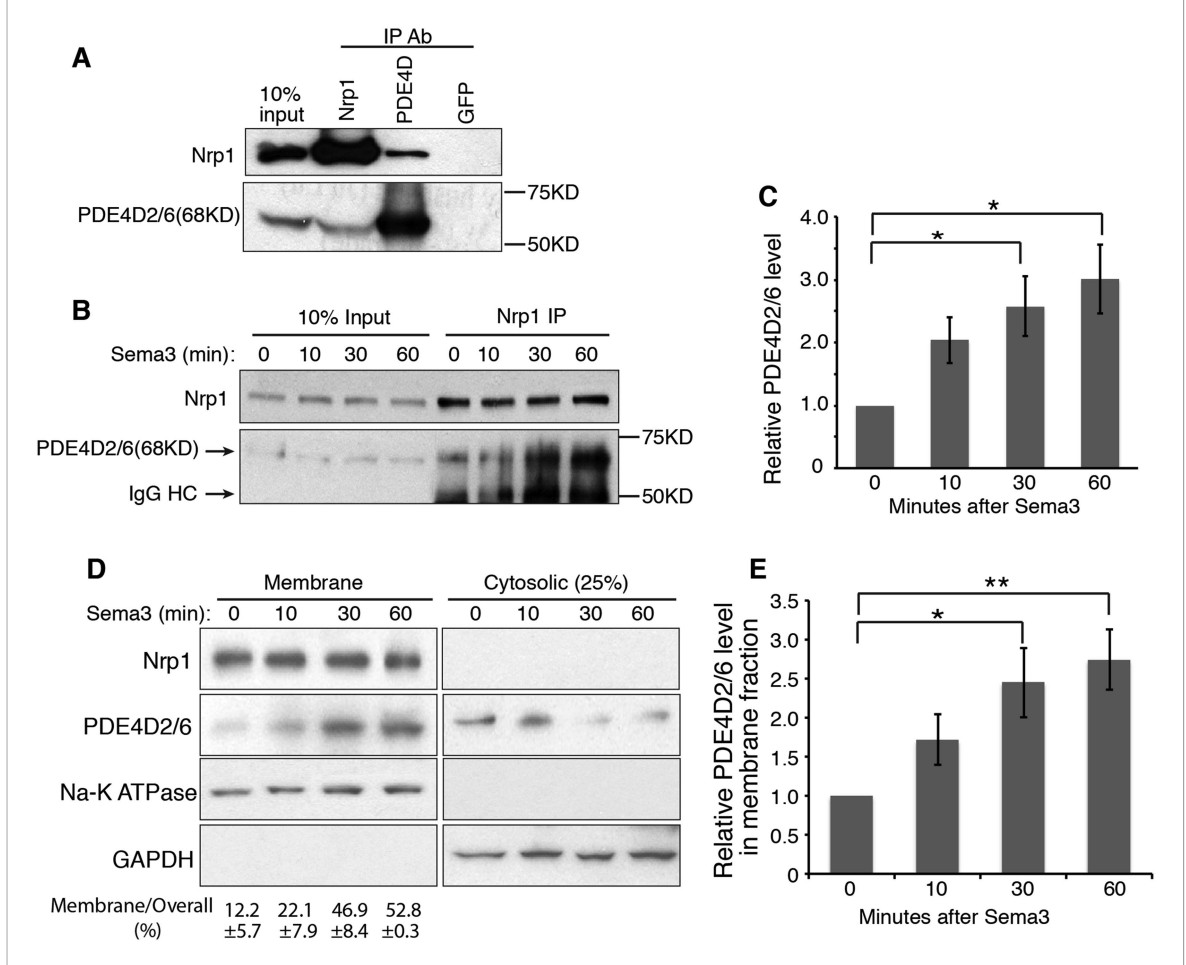

**Figure 2**. Nrp1 interacts with PDE4D, and Sema3 promotes this interaction. (**A**) Immunoblot showing that the endogenous PDE4D isoform 2/6 from NIH3T3 cells was immunoprecipitated by Nrp1 antibody, but not by an unrelated GFP antibody. (**B**, **C**) Immunoblots showing the amount of PDE4D2/6 immunoprecipitated by Nrp1 antibody from cells treated with Sema3 (a combination of Sema3A+3F) for different periods of time. (**B**). Band intensity was normalized to the input at the corresponding time point, and then normalized to time 0 (**C**). (**D**, **E**) Immunoblots showing the amount of Nrp1 and PDE4D2/6 in the membrane and cytosolic fractions from cells treated with Sema3A+3F for different periods of time. (**D**). Band intensity was normalized to time 0 (**E**). Quantification in (**C**) and (**E**) were from three independent experiments. Error bars represent SEM. Statistics: Student's t-Test. *p < 0.05, **p < 0.01.

The following figure supplements are available for figure 2:

**Figure supplement 1**. The cytoplasmic domain of Nrps mediates their interactions with PDE4D2.

**Figure supplement 2**. Sema3-Nrp signaling reduces intracellular cAMP levels.

not recruit PDE4D to the membrane, nor did it increase Sema3's effect on PDE4D membrane translocation (*Figure 2—figure supplement 1D,E*). We then determined whether Nrps are required for Sema3's effect on PDE4D membrane translocation. We silenced both Nrps with lentivirus-mediated RNAi, and found that Sema3 could not promote PDE4D membrane association without Nrps (*Figure 2—figure supplement 1F,G*). We concluded that Sema3 signals through Nrps to promote PDE4D translocation to the cell membrane.

PDE4D hydrolyzes cAMP, which is produced by adenylate cyclase (AC) at the plasma membrane (*Beavo and Brunton, 2002*). The recruitment of PDE4D by Sema3-Nrps to the plasma membrane may bring it close to the site of cAMP production, where it can efficiently degrade cAMP. To test this hypothesis, we measured the change of intracellular cAMP level using a cell-based assay (*Kumar et al., 2007*; *Dunn et al., 2013*; *Kokkinaki et al., 2013*). Sema3F significantly reduced intracellular

cAMP levels compared to a BSA control (*Figure 2—figure supplement 2B*). When we silenced Nrps expression using lentivirus-mediated RNAi, the intracellular cAMP level significantly increased, and Sema3F failed to reduce it (*Figure 2—figure supplement 2C*). In conclusion, Sema3 signals through Nrps to reduce intracellular cAMP levels.

## Sema3-Nrp signaling inhibits PKA activity

cAMP activates multiple protein effectors in the cell, such as PKA, Epac, and cyclic nucleotide-gated ion channels (*Bradley et al., 2005*; *Borland et al., 2009*). PKA is a well-known negative regulator of the Hh pathway. Genetic removal of PKA leads to full activation of the Hh pathway in the developing neural tube (*Epstein et al., 1996*; *Tuson et al., 2011*). The lowering of intracellular cAMP by Sema3F-Nrp signaling has the potential to inhibit PKA, so we examined PKA activity in NIH3T3 cells. Previous publications suggest that phosphorylation of PKA at T197 is essential for PKA activity (*Adams et al., 1995*; *Steichen et al., 2010*), and that the level of phosphorylation of PKA at T197 correlates with PKA activation (*Moore et al., 2002*; *Barzi et al., 2010*). Thus, we used an antibody that specifically recognizes the phosphorylation of PKA at T197 (pPKA-T197) and performed immunofluorescence. In agreement with previous reports (*Barzi et al., 2010*; *Tuson et al., 2011*), phospho-PKA-T197 accumulates at the centrosome, and is distributed throughout the entire cytoplasm in a punctate pattern (*Figure 3A*). Sema3F treatment for 30 min dramatically reduced the pPKA-T197 level in the cytoplasm (*Figure 3B*). The pool of PKA at the cilium base, marked by Pericentrin antibody, also went down (*Figure 3C*); this pool of PKA has been hypothesized to directly participate in Hh regulation (*Barzi et al., 2010*; *Tuson et al., 2011*). To determine whether the reduction in pPKA-T197 level is due to decreased total PKA, we stained the cells with an antibody that detects total PKA. Sema3F did not induce any change in total PKA level in the cytoplasm or at the centrosome (*Figure 3—figure supplement 1A–C*). Therefore, Sema3F reduced the detectable level of PKA phosphorylation at T197. This effect was dependent on the presence of Nrps, since Sema3F did not reduce pPKA-T197 after *Nrp* expression was silenced by lentivirus-mediated RNAi (*Figure 3D–E*), and the total PKA level remained unchanged (*Figure 3—figure supplement 1D,E*).

To further investigate the effect of Sema3 on PKA activity, we detected the pPKA-T197 level using protein blots of NIH3T3 cell lysates. When cells were treated with forskolin, a drug that elevates intracellular cAMP levels, the detectable level of pPKA-T197 increased by 50% (*Figure 3F*). This result suggests that the detectable level of pPKA-T197 is a valid measurement for inferring the PKA activity level. Sema3F treatment reduced the detectable level of pPKA-T197 by ~20% (*Figure 3F*). Next, we further evaluated PKA activity by examining PKA substrate phosphorylation using an antibody that recognizes the phosphorylated PKA consensus motif (Rxx[S/T]P). Substrate phosphorylation level increased by ~130% upon forskolin treatment, and decreased by ~27% when treated with Sema3F (*Figure 3G*).

The results of the experiments using pPKA-T197 measurements suggest that Sema3F acts through Nrps to reduce PKA activity, thus stimulating Hh transduction.

## Regulation of Gli2 and Gli3 by Sema3-Nrp-mediated inhibition of PKA activity

The transcription factors Gli2 and Gli3 are known PKA targets in Hh transduction. Both Gli2 and Gli3 can be phosphorylated by PKA, but the effects of PKA differ qualitatively and quantitatively with respect to the two Gli proteins. PKA-driven phosphorylation leads to proteolytic processing of Gli3 into Gli3R, a potent transcription repressor of Hh target genes (*Humke et al., 2010*; *Tukachinsky et al., 2010*). PKA has a distinct effect on Gli2, affecting proteolysis only very slightly but more dramatically controlling its accumulation at cilia tips, a step that is required for Gli2 activation (*Pan et al., 2006*; *Hui and Angers, 2011*; *Tuson et al., 2011*; *Niewiadomski et al., 2013*).

We evaluated the effect of Sema3-Nrp signaling on Gli3 proteolytic processing using protein Blots. After silencing *Nrps* by lentivirus-mediated RNAi, we did not detect any obvious change in Gli3R production in response to Shh treatment (*Figure 3—figure supplement 2C–F*).

We examined the effect of Sema3-Nrp signaling on Gli2 enrichment at the cilia tips. Consistent with previous reports (*Tukachinsky et al., 2010*; *Niewiadomski et al., 2013*), Shh increased Gli2 levels at cilia tips within a few hours. However, after the expression of both *Nrp* genes was silenced by

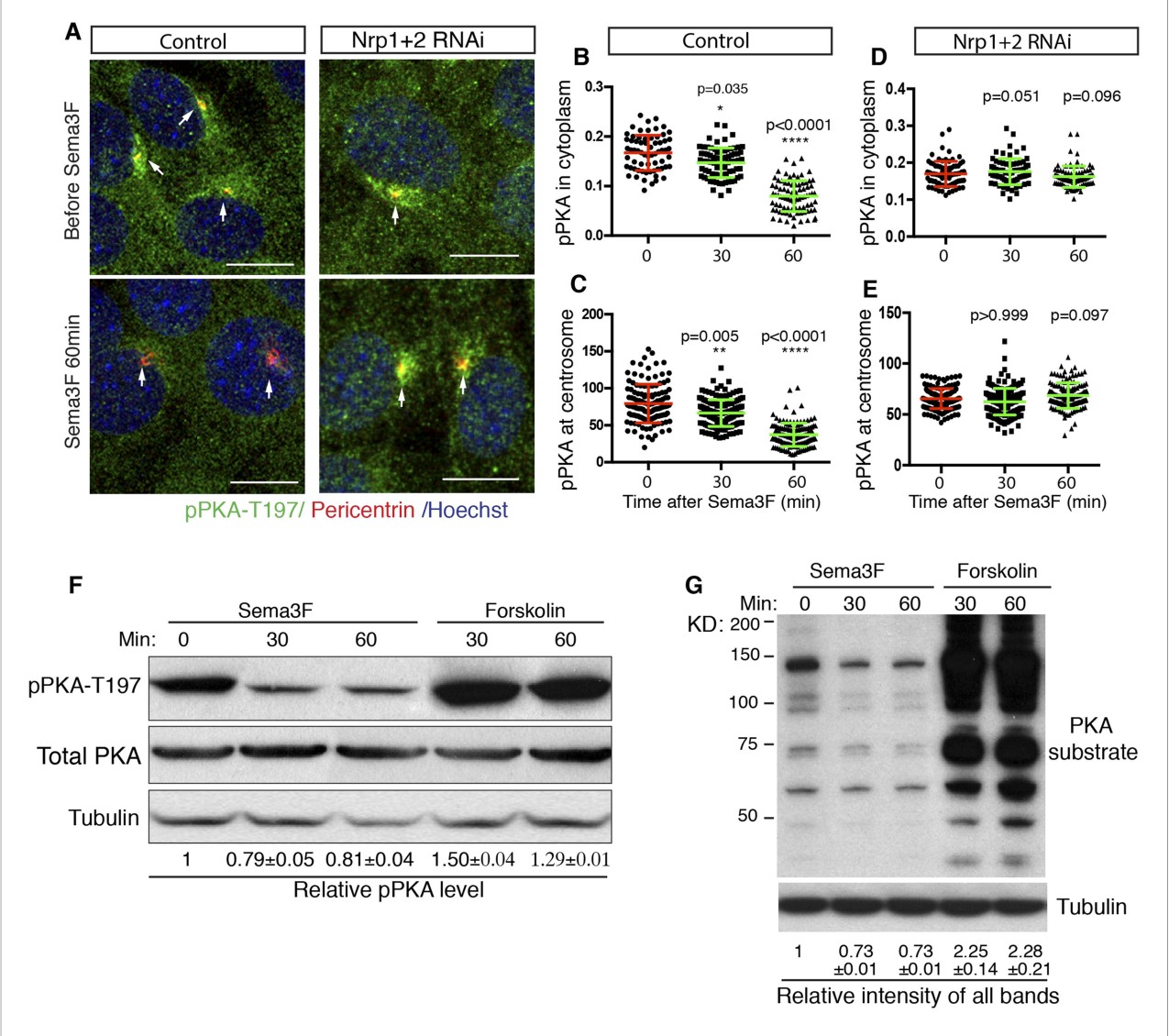

**Figure 3**. Sema3-Nrp signaling inhibits PKA activity. (**A**) Immunofluorescence showing active PKA in NIH3T3 cells after Sema3F treatment. Cells were infected with lentiviruses expressing control shRNA or shRNA against Nrp1&2 for 72 hr. Active PKA is recognized by the antibody against phospho-PKA T197 (green). Scale bar, 10 μm. (**B–E**) Quantification of phospho-PKA level at the cytoplasm and centrosome. Data are shown as mean ± SD. Statistics: Kruskal–Wallis non-parametric One-Way ANOVA. (**F**) Western blot showing the phospho-PKA and total PKA levels in NIH3T3 cells treated with Sema3F or Forskolin for indicated time periods. Quantification of relative active PKA level (mean ± SEM) from three independent experiments was shown at the bottom of each blot. (**G**) Western blot showing the levels of PKA substrate phosphorylation in NIH3T3 cells treated with Sema3F or Forskolin for indicated time periods. Quantification of the relative intensity of all bands in each lane (mean ± SEM) from three independent experiments was shown at the bottom.

The following figure supplements are available for figure 3:

**Figure supplement 1**. Sema3-Nrp signaling does not change total PKA level.

**Figure supplement 2**. Sema3-PKA signaling enhances Shh-induced Gli2 enrichment to the cilia tip, but does not affect Gli3 processing.

RNAi, this increase was significantly attenuated when compared with control RNAi treated cells (*Figure 3—figure supplement 2A,B*). We conclude that Sema3/Nrp-mediated PKA inhibition promotes Hh transduciton by facilitating Gli2 activation, but leaves the level of Gli3 repressor unaffected.

## Inhibition of PDE4D suppresses Hh signal transduction

Our data suggest that signaling downstream of Sema3-Nrp promotes PDE4D activity to inhibit PKA, which eventually enhances Hh transduction (*Figure 4A*). This model indicates that inhibiting PDE4D will suppress Hh transduction. To test this idea we treated cells with rolipram, which inhibits all PDE4 families; GEBR-7b, which specifically inhibits the PDE4D subfamily (*Bruno et al., 2011*); or roflumilast, which targets all PDE4 proteins and is approved by the FDA for treating inflammatory lung disease. Shh was added simultaneously with each individual drug to the cells, and *Gli1* transcription levels were measured to assess Hh transduction. All three drugs inhibited Shh- or SAG-induced Hh target gene activation in a dose-dependent manner (*Figure 4B*, *Figure 4—figure supplement 1B*). To exclude the possibility that PDE4D inhibitors are toxic to the cell, we assessed Wnt signaling by measuring transcripts of the Wnt target gene *Axin2*. Roflumilast and GEBR-7b had no effect on Wnt3a-induced *Axin2* transcription at all concentrations tested, nor did rolipram at concentration lower than 1uM (*Figure 4C*).

We then used shRNA to silence the expression of PDE4D in NIH3T3 cells, and assessed Hh signal transduction. Without PDE4D, the Shh-induced Hh signal transduction was significantly reduced (*Figure 4—figure supplement 2*). These data are fully consistent with a view of PDE4D as a positive regulator of Hh transduction. The SAG result indicates that PDE4D operates in the Hh pathway at the level of Smo or downstream.

## GNP proliferation is promoted by Sema3-Nrp, and inhibited by high PKA activity

MB derives from over-proliferation of GNPs in the developing cerebellum (*Barakat et al., 2010*), and Shh is a strong mitogen that stimulates GNP proliferation (*Dahmane and Ruiz i Altaba, 1999*; *Wallace, 1999*;

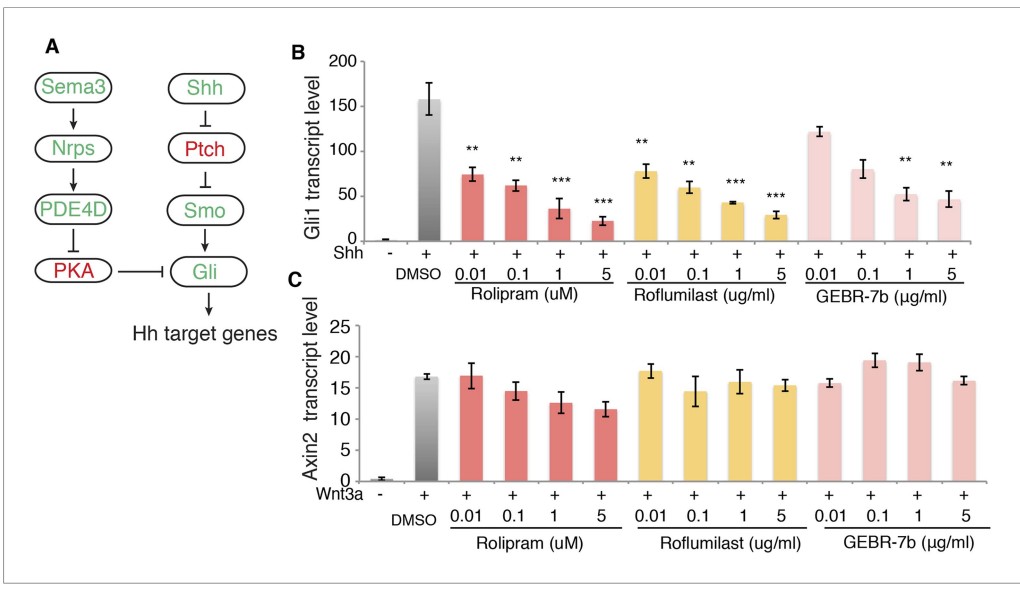

**Figure 4**. PDE4D inhibitors suppress Hh signal transduction. (**A**) Schematic diagram showing that Sema3-Nrp signaling regulates Hh pathway through controlling the activity of PDE4D and PKA. (**B**) The Hh signaling activity in NIH3T3 cells incubated with PDE4D inhibitors together with Shh conditioned medium for 4 hr *Gli1* transcript levels were measured by qPCR to evaluate the Hh signal transduction. (**C**) The Wnt signaling activity in NIH3T3 cells incubated with PDE4D inhibitors together with Wnt3a for 4 hr *Axin2* transcript levels were measured by qPCR to evaluate the Wnt signal transduction. All data are mean ± SEM. Statistics: Student t-Test, in comparison to the condition where cells were treated with DMSO. **p < 0.01, ***p < 0.001.

The following figure supplements are available for figure 4:

**Figure supplement 1**. PDE4D inhibitors suppress SAG-induced Hh signal transduction.

**Figure supplement 2**. PDE4D knockdown reduces Hh signaling activity.

*Wechsler-Reya and Scott, 1999*). To test whether Sema3-Nrp-PDE4D signaling controls GNP proliferation, we cultured GNPs in dishes and evaluated their proliferation with BrdU incorporation assay (*Figure 5A*). The BrdU incorporation rate was doubled by SAG, and further increased by Sema3F. The effect of Sema3F was blocked by the Nrp[panA] antibody, which interferes with Sema3-Nrp interaction, but not by the Nrp[2B] antibody, which blocks VEGF. Activating PKA with forskolin or inhibiting PDE4D with roflumilast completely abolished the Sema3F effect and reduced GNP proliferation to the baseline level (*Figure 5B*). Thus the molecular mechanisms linking Sema3-Nrp signaling with the Hh pathway control GNP proliferation.

## Genetic removal of Sema3-Nrp signaling impairs GNP proliferation in the developing cerebellum

After establishing the molecular mechanism that integrates Nrp and Hh signaling in vitro, we explored the interplay of the two signaling pathways in vivo. To this end, we genetically removed both *Nrp1* and *Nrp2*. To circumvent the early lethality of *Nrp1* knockout mice (*Kawasaki et al., 1999*), we introduced a *Math1-Cre* allele into *Nrp1[floxed/floxed]*, *Nrp2[−/−]* mice (*Giger et al., 2000*; *Gu et al., 2003*). The *Math1* enhancer drives Cre production in GNPs early in development (*Matei et al., 2005*). Immunofluorescent staining of P7 cerebellum showed that *Nrp1* is expressed in the external granule cell (EGL) layer, which contains GNP cell bodies, in the molecular layer (ML), which contains parallel fibers of differentiated granule neurons, and in blood vessels (*Figure 6A*). In *Nrp1[floxed/floxed]*, *Nrp2[−/−]*, *Math1-Cre* mice (DKO), Nrp1 fluorescent signals disappeared from EGL and ML, but remained in blood vessels (*Figure 6A*), demonstrating the specific knock-out of *Nrp1* in GNPs. A protein blot of P7 cerebellar lysate from these mice showed elimination of detectable Nrp2 and little to no Nrp1 (*Figure 6B*).

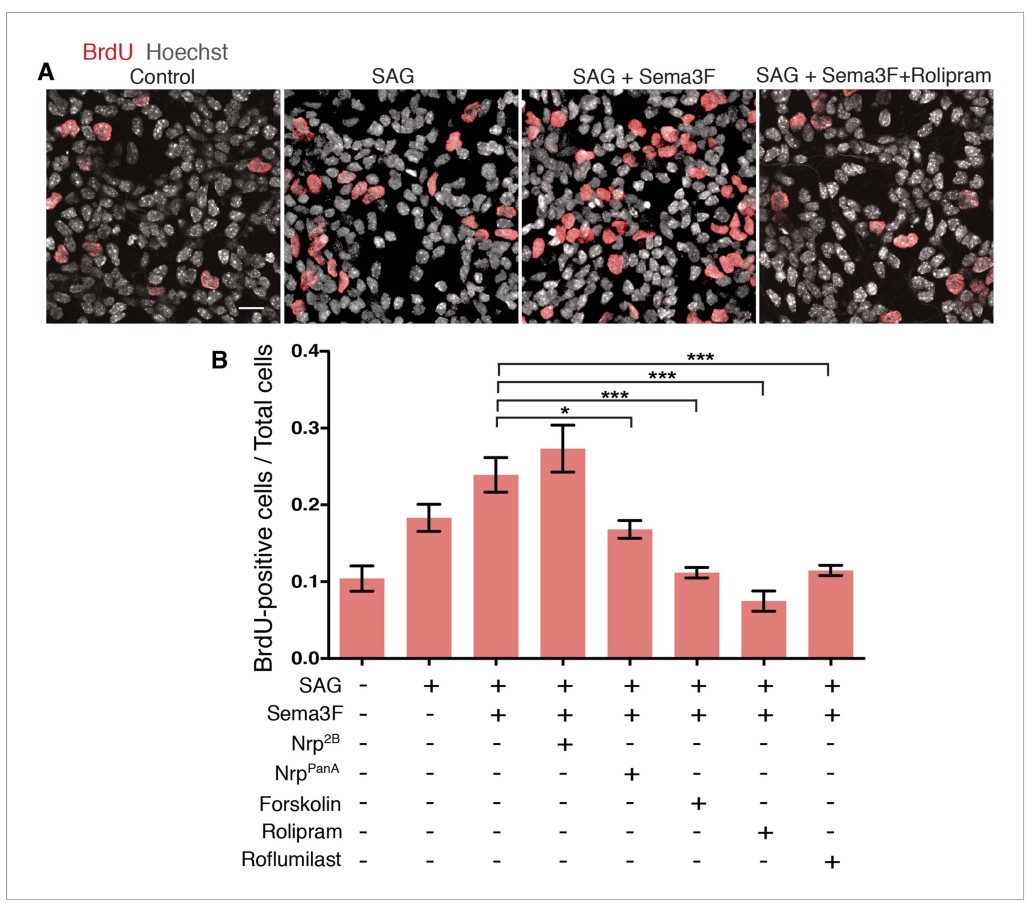

**Figure 5**. GNP proliferation is promoted by Sema3 and inhibited by hyperactive PKA. (**A**) GNP proliferation was assayed by BrdU incorporation in GNPs cultured in vitro for 20 hr. GNPs were cultured in medium with indicated reagents and BrdU was added 1 hr before the cells were fixed. Scale: 20 μm. (**B**) BrdU incorporation rate was calculated as the number of BrdU-positive cells divided by total cell number. Error bars represent SEM, Statistics: Kruskal–Wallis non-parametric One-Way ANOVA. *p < 0.05, ***p < 0.001.

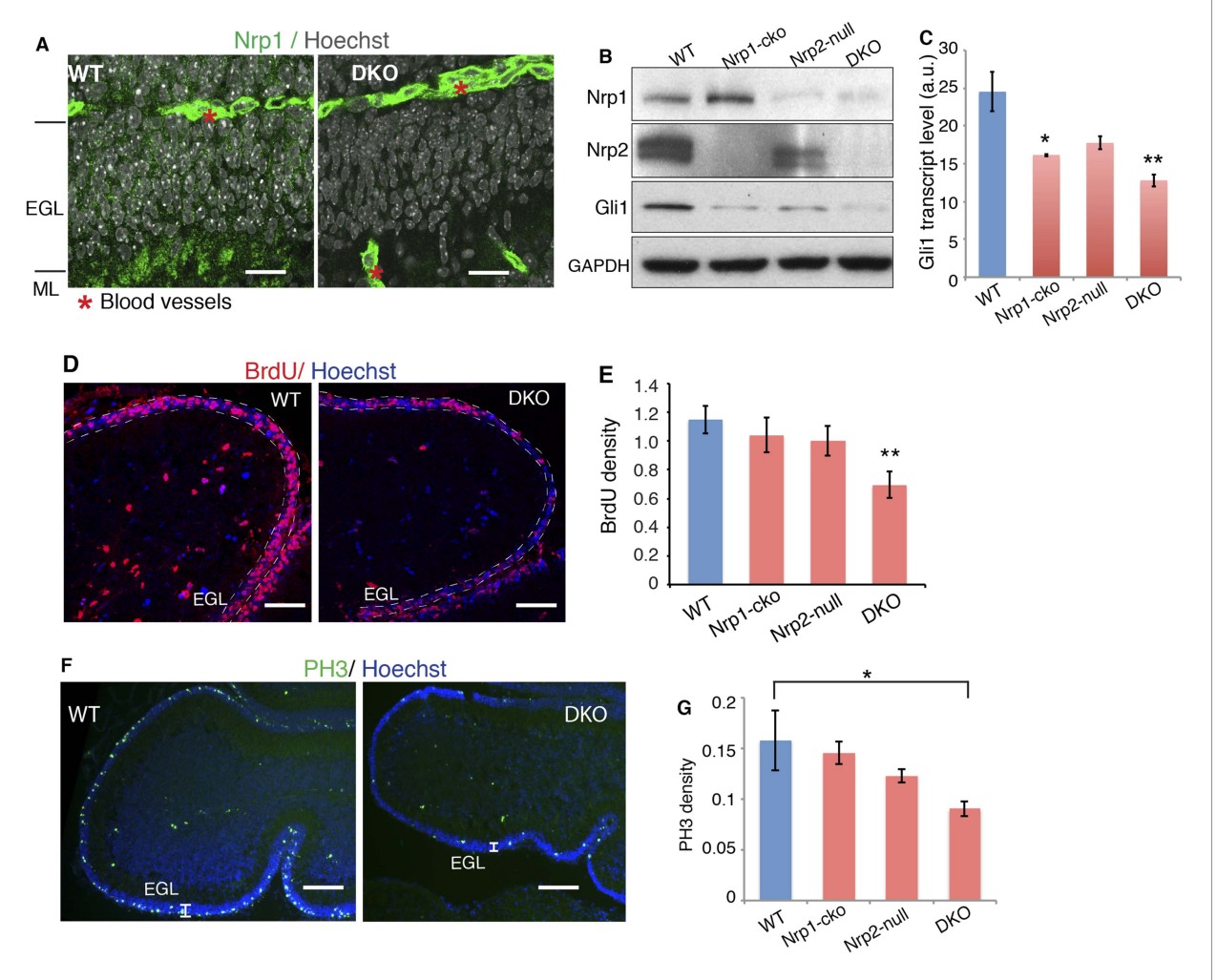

**Figure 6**. Loss of Nrps impairs GNP proliferation in the developing cerebellum. (**A**) Immunostaining of Nrp1 (green) in the P7 cerebellum of wild type and *Nrp1/2* DKO mouse show that Nrp1 was specifically knocked out from GNP cells in EGL and parallel fibers in ML, but remains in blood vessels. Scale, 20 μm. (**B**) Immunoblots showing that Nrp1 and 2 are knocked out from DKO cerebellum, and that Gli1 expression was much lower in the cerebellum of Nrp1/2 DKO mice compared to wild type littermates. Nrp1-cko: *Math1Cre,Nrp1^{floxed/floxed}*. Nrp2-null: *Nrp2^{-/-}*. (**C**) The Hh pathway activity in the cerebellum of Nrp1/2 DKO mice and littermates was evaluated by *Gli1* transcript level. (**D, E**) GNP proliferation in the EGL (circled by dotted lines) of Nrp1/2 DKO mice and littermates was evaluated by BrdU incorporation assay. Scale: 50 μm. (**F, G**) Immunostaining and quantification of Phospho-H3 (green) in EGL of wild type and Nrp1/2 DKO cerebellum at P7. Scale, 100 μm. All error bars represent SEM. Statistics: non-parametric Mann–Whitney test. *p < 0.05, **p < 0.01.

The following figure supplement is available for figure 6:

**Figure supplement 1**. Loss of Nrps in the developing cerebellum leads to PKA hyperactivation.

Hh transduction was severely impaired in GNPs in *Nrp* double knock-out (DKO) mice, as assessed by *Gli1* transcript and protein levels (***Figure 6B,C***). BrdU incorporation was used to assess GNP proliferation. BrdU density was much lower in the EGL of DKO mice, reflecting GNP proliferation defects (***Figure 6D,E***). The density of mitotic cells indicated by mitotic marker PH3 also significantly decreased in DKO mice (***Figure 6F,G***). Thus genetically eliminating Sema3-Nrp signaling compromises Hh transduction in vivo, and consequently impairs GNP proliferation.

We examined PKA activity in cerebella of *Nrp* DKO mice. In agreement with the results from NIH3T3 cells, PKA activity, assessed with pPKA-T197 antibody, was increased by 1.4 fold in the DKO, whereas the total PKA level remained unchanged (***Figure 6F—figure supplement 1A***). Blots with the antibody that recognizes the phosphorylated PKA consensus motif showed that phosphorylation of PKA

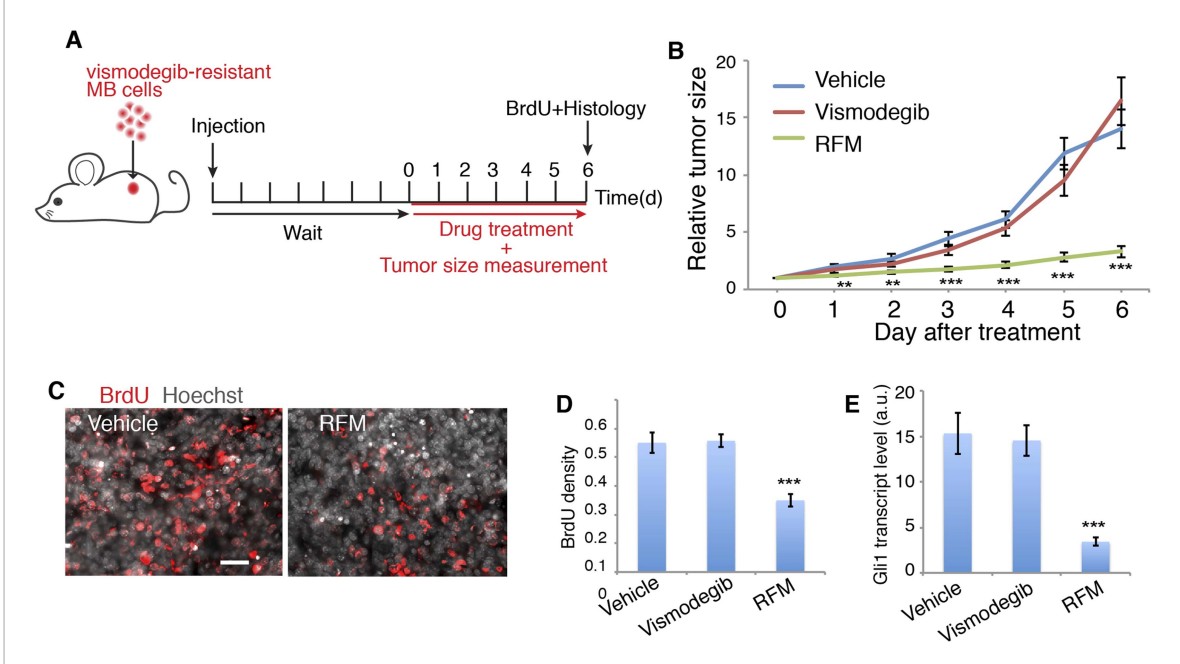

**Figure 7.** PDE4 inhibitors suppress Hh signal transduction and Hh-related tumor growth. (**A**) Schematic diagram of the MB tumor allograft experiment in mouse. Drug treatment started 6 day after injection when the size of tumors could be accurately measured. (**B**) The relative tumor size is defined as the tumor volume on the indicated day divided by that on day 0. For each drug 9–10 mice were used. (**C**, **D**) BrdU density in tumors after drug treatment. Scale: 50 μm. (**E**) Hh signal transduction in tumors was assessed by *Gli1* transcription level through qPCR. RFM: roflumilast. Error bars represent SEM. Statistics: Student's t-Test. **p < 0.01, ***p < 0.001.

The following figure supplement is available for figure 7:

**Figure supplement 1.** Schematic view of the integration of Sema3-Nrp with the Hh signal pathway by controlling PKA and PDE4.

substrates was greatly increased in DKO cerebellum (*Figure 6—figure supplement 1B*). In conclusion, eliminating Sema3-Nrp signaling from the cerebellum increases PKA activity.

## PDE4D inhibitor blocks growth of Hh-related MB that is resistant to vismodegib

Next, we tested the effect of blocking Sema3-Nrp-PDE4D signaling on MB growth. We targeted PDE4D because roflumilast, the FDA-approved PDE4D inhibitor, blocks Hh transduction (*Figure 4*) and is readily available for patients. Since PDE4D regulates Hh transduction downstream of Smo, roflumilast would be an ideal drug to treat vismodegib-resistant tumors. We tested this hypothesis in mouse MB tumors containing the Smo mutation (Smo[D477G]) that blocks vismodegib binding with Smo (*Kim et al., 2013*) (*Figure 7A*). In mouse MB allografts, roflumilast dramatically inhibited tumor growth starting from the first day after treatment. In contrast, vismodegib had no effect on tumor growth, as expected (*Figure 7B,C*). Roflumilast treatment also significantly reduced cell proliferation in tumors as evaluated by BrdU incorporation (*Figure 7C*). Hh signal transduction dramatically decreased, as assessed by *Gli1* transcript level in tumors (*Figure 7E*). Thus roflumilast inhibited Hh signal transduction and suppressed the growth of vismodegib-resistant tumors.

## Discussion

### Sema3-Nrp pathway enhances Hh signal transduction during development and disease

Hh and Sema3-Nrp signaling are simultaneously involved in many developmental processes and diseases. The interaction between the two pathways was revealed only recently by our previous

study (*Hillman et al., 2011*). We found that Nrps positively regulate Hh transduction, and Hh in turn induces *Nrp1* expression. In this previous study, we excluded the possibility that Nrps regulate binding or processing of the Hh ligand, and pinpointed Nrp participation in the Hh pathway at steps between Smo and SuFu. Beyond these findings, how the two pathways interact at the molecular level has been unknown. In the present study we discovered that the Sema3-Nrp pathway integrates with the Hh pathway by controlling the subcellular localization of PDE4D and consequently cAMP levels and PKA activity. Sema3 enhances the recruitment of PDE4D to the cell membrane by promoting its binding to the Nrp cytoplasmic domain (*Figure 7—figure supplement 1*). As most cAMP is produced by AC at the cell membrane, we speculate that the recruitment of PDE4D brings the enzyme into close proximity to the site of cAMP production and may enable more efficient hydrolysis of synthesized cAMP before it diffuses throughout the entire cytoplasm, including the cilium base. Consequently PKA is inhibited, which in turn promotes Hh transduction.

Sema3 by itself cannot activate Hh signal transduction (*Figure 1C*), but does enhance Hh signaling that has been activated by Shh or SAG (*Figure 1C*, *Figure1—figure supplement 1A,B*). Therefore, we believe that the Sema3-Nrp signaling is modulatory rather than permissive for the Hh pathway, but nevertheless is a powerful facilitator of Hh transduction. Sema3-Nrp signaling is likely to contribute to Hh transduction in many tissues in vivo. During the development of many tissues, *Sema3* genes and *Shh* are co-expressed, and the Hh pathway functions together with Nrps. For example, in the developing neural tube Sema3 and Shh act together to control the midline crossing of commissural axons (*Parra and Zou, 2010*). In developing skin where Hh signaling is critical for hair follicle development, Nrps are co-expressed in hair follicle cells with Hh signaling components such as Smo (*Hillman et al., 2011*). In developing cerebellum *Sema3* genes are co-expressed with *Shh* in Purkinje neurons. The two kinds of ligands may be released together to stimulate proliferation of GNPs in the cerebellar external granule layer (http://www.gensat.org/imagenavigator.jsp?imageID=50656). Indeed, we found that genetic knockout of both *Nrp* genes blocks Sema3 signaling and impairs GNP proliferation in vivo. Finally, RNA-seq results showed that in pediatric medulloblastoma, Sema3 and Nrp1 expression levels are significantly elevated (*Cho et al., 2011*). This elevated expression may contribute to increased Hh signaling activity that promotes tumor growth.

## Nrps are promising therapeutic targets for Hh-related tumors

Nrps are known as multi-functional co-receptors that bind to different ligands in different cell types (*Gu et al., 2003*; *Gitler et al., 2004*; *Pellet-Many et al., 2008*). Sema3 and VEGF family proteins, both ligands for Nrp receptor complexes, activate distinct transduction cascades. At least three pathways lie downstream of Nrps. First, Sema3 activates a small GTPase of the Rho family, and regulates the reorganization of actin cytoskeleton in neurons. This pathway is implicated in growth cone collapse (*Chen et al., 1998*; *Polleux et al., 1998*; *Zanata et al., 2002*). Second, VEGF family proteins signal through phospholipase C (PLC) to activate the pro-survival/pro-growth factor ERK1/2 and PI3K/Akt, and contribute to endothelial stimulation and pathologic angiogenesis (*Soker et al., 1998*; *Bielenberg et al., 2006*). Third, Sema3 signaling reduces PKA activity in growing axons via unknown mechanisms (*Parra and Zou, 2010*; *Shelly et al., 2011*).

The signaling mechanism downstream of Nrps that contributes to MB growth also involves these signaling pathways. *Snuderl et al. (2013)* found that the placental growth factor (PLGF), a member of the VEGF family, acts through Nrp1 to promote MB growth. Their study suggested that the PLGF-Nrp1 signaling is involved in all subgroups of MB. At the molecular level, PLGF does so by activating ERK1/2 and PI3K/Akt. In our study, we found that Sema3 signals through Nrps to promote PDE4D activity and inhibit PKA, which ultimately potentiates Hh target gene activation. The Sema3-Nrp-PDE4D signaling is involved in the Shh subgroup of MB. Although the two studies revealed different signaling mechanisms downstream of Nrps focusing on different subgroups of MBs, both studies highlight the involvement of Nrps in the development of MB, and highlight Nrp1 as a promising therapeutic target for this devastating brain tumor.

## The Sema3-Nrp-PDE4D pathway provides a new regulatory mechanism of PKA in Hh transduction

It has been proposed that cAMP-PKA activity at the base of cilia is crucial in regulating Hh signal transduction. The concentration of cAMP is controlled by two enzymes: the AC that produces cAMP,

and the phosphsdiesterase (PDE) that hydrolyzes it. A recent study (*Mukhopadhyay et al., 2013*) reported a mechanism through which GPR161 activates Gαs-AC to produce cAMP at the cilia base, thereby switching off Hh signal transduction. Shh triggers the internalization of GPR161 from cilia, which switches on Hh signaling. This landmark work provides mechanistic insights into the coupling of cAMP-PKA activity to the on and off switch of the Hh pathway. Our study revealed the mechanism of cAMP control in Hh pathway—PDE hydrolysis. Furthermore, we found that Sema3-Nrp signal-mediated cAMP degradation does not switch on Hh transduction, but rather potentiates already activated Hh transduction. We believe that the mechanism of cAMP control by the GPR161-Gαs-AC and by Sema3-Nrp-PDE4D signaling coexist in the same cell, and may act at different stages of Hh signaling: GPR161 as a switch, and Sema3-Nrp as an amplifier.

The substrates of PKA on Hh pathway are Gli transcription factors. In the absence of Hh ligand, PKA phosphorylates Gli3 and subsequently Gli3 is proteolytically processed into Gli3 repressor (*Wang et al., 2000*). Gli2 is the major Hh pathway transcriptional activator; its activation is controlled by translocation to the cilium (*Humke et al., 2010*; *Tukachinsky et al., 2010*). From our experiments we conclude that Sema3-Nrp signaling inhibits PKA and that inhibition of *Nrp* functions impairs Shh-induced Gli2 enrichment at cilia tips. These observations support the idea that overactive PKA following Nrp RNAi is the mechanism of reduced Gli2 enrichment at cilia tips. This result is consistent with two previous findings. First, activation of PKA by Forskolin inhibits cilium translocation of the SuFu–Gli2/3 complex (*Humke et al., 2010*; *Tukachinsky et al., 2010*). Second, Gli2 localization to cilia tips is elevated in *PKA*-null mice (*Tuson et al., 2011*). These studies also suggest that, within cilia, the SuFu–Gli complex is dissociated by active Smo, and that only after the dissociation can free Gli2 be activated and translocated into the nucleus to turn on target gene transcription. Therefore, without activation by Hh ligand, Gli2 cannot be active even if it is transported to cilia. This may explain why Sema3-Nrp signaling cannot, on its own, turn on Hh target genes, but does amplify Hh-triggered transcription.

Intriguingly, we found that Sema3-Nrp is not involved in the production of Gli3R (*Figure 3—figure supplement 2C–F*). Our finding is compatible with the existence of a PKA-independent mechanism of Gli3 processing. As shown by *Tuson et al. (2011)*, a significant portion of Gli3 is processed into Gli3R in *PKA*-null mice. This mechanism of Gli3 processing may kick in when PKA activity is inhibited by a Sema3-Nrp signal. Alternatively, PKA may control Gli2 and Gli3 at distinct subcellular locations, and Sema3-Nrp only modulates PKA activity where it controls Gli2 activation. A similar divergence between Gli2 activation and Gli3R regulation has been reported in *Arl13B*-null mice (*Caspary et al., 2007*). In these mice, Gli3R is produced normally, whereas Gli2 is hyperactive. Another example is Eya1, a phosphatase that positively regulates Hh signaling. *Eya1* mutants exhibited reduced Hh transduction, but Gli3 processing was normal (*Eisner et al., 2015*). These findings suggest that Gli2 activation and Gli3 repressor formation are independently regulated during Hh signal transduction.

Finally, a third possibility exists: Gli3 processing and Gli2 inhibition could be sensitive to different levels of PKA activity. In cells without Nrps, Shh could still be able to lower PKA activity at the cilium base by inhibiting GPR161 to block Gli3 processing, whereas residual PKA activity could be sufficient to suppress Gli2 activation. In this model, the Sema3-Nrp-PDE4D pathway we describe here might act in parallel to Shh-Gpr161 pathway to more robustly reduce PKA activity in cilium, leading to more robust Gli2 activation.

## Sema3-Nrp signal controls PDE4D localization to modulate Hh signal transduction

Nrps typically form complexes with the co-receptors Plexin and VEGF receptor to transduce signals from Sema3 and VEGF. However, accumulating evidence suggest that Nrps support intracellular signaling independent of their known co-receptors (*Pellet-Many et al., 2008*). The short cytoplasmic domain of Nrp1 has been reported to bind to a PDZ-domain containing protein GIPC1 (*Cai and Reed, 1999*). We could not detect an interaction between Nrp1 and GIPC1 in immunoprecipitation assays (data not shown), perhaps due to the different cell type used in our experiment.

We discovered another protein that interacts with the Nrp1 cytoplasmic domain: PDE4D. Sema3 enhances the Nrp-PDE4D interaction and promotes the recruitment of PDE4D to the plasma membrane. Subcellular localization of PDE4D is an important mechanism for the precise control of cAMP concentration in the cell (*Houslay, 2010*). In airway epithelium PDE4D localizes to the apical

domains of cells, where it degrades cAMP and prevents cAMP diffusion into other cellular locations (*Barnes et al., 2005*). In cardiac myocytes PDE4D is rapidly recruited to the cell membrane by activated β2-adrenergic receptor to precisely control the spatial and temporal cAMP concentration (*Perry et al., 2002*). These previous findings are consistent with our proposed model in which PDE4D efficiently degrades cAMP at the site of cAMP production, thereby reducing cAMP concentration throughout the cytosol and at the cilium base. We showed that this reduced cAMP concentration inhibited PKA activity, thus promoting Hh transduction. Our discovery was confirmed by a recent study in zebrafish showing that a small molecule that inhibits PDE4 promotes Hh signal transduction (*Williams et al., 2015*). In additional to PKA, cAMP activates EPAC proteins, guanine nucleotide exchange factors for Ras-like GTPases (*Borland et al., 2009*), and cyclic nucleotide-gated ion channels in the cilia of the olfactory neurons (*Bradley et al., 2005*). It will be intriguing to see whether these cAMP targets also contribute to Hh transduction or other Hh-triggered effects on cells.

### PDE4D inhibitors are potential drugs for treating Hh-related tumors

Recently genome wide sequencing studies implicated mutations in PDE4D and PKA in adult Hh-related MB (*Kool et al., 2014*), independently suggesting PDE4D and PKA as potential therapeutic targets. Our studies elucidate a mechanism that may well underlie these genetic phenomena. Remarkably, we found that targeting PDE4D to inhibit the newly discovered Sema3-Nrp-PDE4D-PKA pathway powerfully blocks the growth of Hh-related MBs that are resistant to Smo inhibitors. One of the PDE4D inhibitors, roflumilast, is used in clinics to treat chronic obstructive pulmonary disease (COPD) (*Giembycz and Maurice, 2014*). In addition to roflumilast, many PDE4D inhibitors are under development to improve cognitive and psychological performances, some of which are in clinical trials (*Gavalda and Roberts, 2013*). Our findings highlight the priority of repurposing PDE4D inhibitors as solo or combination therapies for Hh-related MB.

## Materials and methods

### DNA constructs and reagents

Human Nrp1 and Nrp2 cDNA were subcloned into pEGFP-N1 vector. Human PDE4D2 cDNA and a triplicated Flag (3×Flag) sequence (DYKDHDG-DYKDHDI-DYKDDDDK) were cloned into the pPBbrs2 vector, using Gibson assembly to place the 3×Flag tag at the C-terminus of PDE4D2. ShhN conditional medium was obtained using a HEK 293 cell line that stably secretes ShhN. Recombinant Shh, Wnt3a and all Sema3 proteins were obtained from R&D Systems (MN, USA), and reconstituted according to the instructions of the manufacture. SAG, Forskolin, Rolipram, Roflumilast and GEBR-7b were obtained from Millipore (NY, USA).

### Mouse strains

Mouse strains of *Nrp1* conditional knockout (*Nrp1^f/f^*), *Nrp2^−/−^*, and *Math1-Cre* were described previously (*Gu et al., 2003*; *Matei et al., 2005*).

### GNP Isolation from cerebellum and drug treatment

GNPs were isolated from P7 cerebellum using a protocol modified from *Wechsler-Reya and Scott (1999)*. Briefly, cerebella from P7 mice were removed, cut into small pieces with razor blades, and incubated at 37°C for 30 min in digestion buffer consisting of 1x HBSS (ThermoFisher, NY, United States, 14,185) with 10 U/ml papain (Worthington, NJ, United States, LSOO3126) and 250 U/ml DNase (Sigma, MO, United States, D4627). At the end of the incubation, the digestion buffer was removed and replaced with 1x HBSS containing 8 mg/ml Ovomucoid (Worthington, NJ, United States, LK003182), 8 mg/ml bovine serum albumin (Sigma), and 250 U/ml DNase. Tissues were then triturated using Pasteur pipettes to obtain a single-cell suspension. Cells were centrifuged at room temperature and resuspended in PBS containing 0.02% BSA (PBS/BSA). The cell suspension was passed through a cell strainer (VWR, 21,008-952) to remove debris. The cell suspension was then underlaid with a step gradient of 35% and 65% Percoll (Sigma, P4937) and centrifuged at high speed for 12 min at room temperature. Granule neuron precursors were harvested from the 35/65% interface and washed in PBS/BSA. Cells were then resuspended in Neurobasal medium (Life Technologies, CA, United States, 21,103-0449) supplemented with GlutaMax (ThermoFisher, 35,050-061) and Penicillin/streptomycin,

and plated on tissue culture dishes coated with Poly-D-Lysine (Millipore, MA, United States, A-003-E) and Laminin (Sigma, L2020).

Drugs were added at the time when cells were plated. Sema3F(R&D, MN) was used at 3 μg/ml, VEGF165 (R&D, MN) was used at 100 ng/ml, Nrp antibodies (provided by Genentech, CA, United States) were used at 50 μg/ml. SAG: 100 μM; Forskolin: 5 μM; Rolipram: 1 μM; Roflumilast: 1 μg/ml. 2 hr BrdU pulse label was done 24 hr after plating. Cells were fixed with 4% paraformaldehyde, and underwent BrdU immunofluorescence. Mouse anti-BrdU (monoclonal, 1:500, Dakocytomation, CA, United States) was used.

## Cell immunofluorescence and microscopy

NIH3T3 cells were plated 24 hr before each experiment at 70% confluency. The cultures were treated with vehicle (0.1% carrier protein BSA in PBS) or Sema3F (3 μg/ml) for 30 min or 1 hr. Cells were fixed with 4% paraformaldehyde for 10 min at room temperature. For Pericentrin staining, cells were then incubated with pre-cooled methanol at −20 °C for 8 min. After that cells underwent immunofluorescence staining. Primary antibodies were phosphoPKA-T197 (Abcam, MA), PKA Cα (Cell signaling, MA), pericentrin (BD Biosciences, CA), acetylated tubulin (Sigma, MO). Images were taken with an inverted laser-scanning confocal microscope (DMIRE2; Leica).

## Image processing for PKA

Confocal imaging stacks of Z-section scanning were superimposed into one image. The cytoplasmic levels of PhosphoPKA and PKA were analyzed using the open-source software program CellProfiler (Broad Institute, Cambridge, MA). The process had four main steps: identification of nuclei (Primary objects), identification of nuclei plus cytoplasm (Secondary objects), determination of cytoplasm (Tertiary objects), and measurement of signals in tertiary objects. Once the nuclei were identified, the secondary objects were determined by dilating primary objects by 15 pixels. The cytoplasm was identified by subtracting the nuclei from the secondary objects. Finally the mean gray level in the cytoplasm was calculated and exported to a spreadsheet.

The levels of phosphoPKA and PKA in centrosome were measured using a customized program written in Matlab (MathWorks, Natick, MA). First the contour of pericentriolar area was delineated based on the signal intensity in the red channel (pericentrin staining). The program then measured the mean gray value of the enclosed area in the green channel (PhosphoPKA and PKA staining) (V1). The contours of each pericentriolar area were then manually moved to a nearby extracellular region, and the mean gray level of the enclosed area in the green channel was measured as background (V2). The final values of PhosphoPKA and PKA were calculated as V = V1−V2. For each condition, 80–100 cells were measured, and data from one of the three independent experiments were shown.

## Subcellular fractionation and immunoblot

Subcellular fractionation was modified from the protocol in (*Humke et al., 2010, Genes Dev*). Briefly, cells were cross-linked with 0.1 mM DSP at room temperature for 15 min, washed with 100 mM Tris buffer pH 7.5, and scraped off the plate. After centrifugation at 1000×*g*, 10 min, the cell pellet was re-suspended and incubated in hypotonic solution (10 mM HEPES, pH 7.9) for 10 min on ice. Cells were pelleted again at 1000×*g*, and homogenized in SEAT buffer (10 mM Triethanolmine pH 7.4, 10 mM Acetic Acid, 1 mM EDTA, 250 mM Sucrose, Protease Inhibitors) with a Dounce homogenizer. Nuclei were separated by centrifugation at 900×*g* for 5 min, twice, and the supernatant was then spun in an ultracentrifuge at 95,000×*g* for 20 min at 4°C. The supernatant was kept as the cytosolic fraction, and was concentrated 10-fold using an Amicon centrifugal filter (10KD). The pellet was designated as the membrane fraction, and was resuspended in RIPA buffer. For each fraction, the protein concentration was determined using a BCA kit (ThermoFisher), and an equal amount of total protein was loaded into each lane of the gel.

For immunoblot to detect endogenous protein, cells were lysed in RIPA buffer (25 mM Tris pH 7.4, 150 mM NaCl, 1% NP-40, 1% sodium deoxycholate, 0.1% SDS, 1 mM DTT, 1 mM PMSF, 10 mM NaF, 50 mM Na-pyrophosphate, 5 mM EDTA, Roche protease inhibitor cocktail, Roche PhosphoSTOP cocktail) for 30 min at 4°C. The lysate was clarified by centrifugation at 14,000 rpm for 30 min. Protein concentrations of the supernatants were determined using the detergent-insensitive BCA kit (Pierce).

Equal amounts of total protein from the samples were supplemented with 6× SDS sample buffer, and boiled for 5 min. Proteins were resolved using SDS-PAGE, and processed for immunoblotting.

Primary antibodies used for immunoblot: Gli1, Nrp2, GAPDH, phospho- PKA-T197, and PKA substrate are from Cell Signaling(MA); Nrp1 and VEGFR1 were from AbCam (MA). Sema3F is from R&D (MN).

## cAMP-Glo assay to detect intracellular cAMP levels

The intracellular cAMP concentration was measured using the cAMP-Glo assay kit from Promega according to the manufacture's instruction. The cAMP standard curve was generated using purified cAMP, from which the relative intracellular level of cAMP was inferred. For each drug treatment, 3 biological repeats were used, and each experiment was repeated 2–3 times.

## Statistical analyses

All data are expressed as mean ± SEM unless otherwise indicated. For statistical analysis of active and total PKA levels, Prism statistical analysis software was used (GraphPad Software, CA). Kruskal–Wallis non-parametric One-Way ANOVA was used to assess significance of multiple data points. Non-parametric Mann–Whitney test was used between data comparing only two groups. For qPCR, the data were analyzed in Excel with Student t-Test. We consider a p value less than 0.05 statistically significant.

## Acknowledgements

We thank Dr. Ju Lu for writing Matlab scripts for analyzing immunofluorescence data, and Dr. Kyuho Han for his help with CellProfiler. We thank Prof. Rajat Rohatgi and Prof. Peter Jackson for helpful discussions. Dr. Helen Rayburn helped with maintenance of the mouse colony. X.G. is supported by a Walter and Idun Berry Postdoctoral Fellowship. The research was supported by N.I.H. grant R01 GM095948 to M.P.S.

## Additional information

### Funding

| Funder | Grant reference | Author |
|---|---|---|
| National Institute of General Medical Sciences (NIGMS) | R01 grant GM095948 | Matthew P Scott |
| Stanford University (su) | The Walter V. and Idun Berry Postdoctoral Fellowship Program | Xuecai Ge |

The funders had no role in study design, data collection and interpretation, or the decision to submit the work for publication.

### Author contributions

XG, Conception and design, Acquisition of data, Analysis and interpretation of data, Drafting the article; LM, Acquisition of data on GNP proliferation, Analysis and interpretation of data on GNP proliferation; KS, Acquisition of data on Neuropilin staining in mouse tissues; TH, Acquisition of data on Western blot of phosphorylated PKA; TP, Participate acquisition of data on study of tumor allograft; AW, Contributed to data acquisition and analysis of PKA activity, which played an important role in leading to the conclusion about PKA activity; TM, Revising the article, supervising the revision of the article, Contributed unpublished essential data or reagents; MPS, Conception and design, Drafting or revising the article

### Ethics

Animal experimentation: This study was performed in strict accordance with the recommendations in the Guide for the Care and Use of Laboratory Animals of the National Institutes of Health. All of the animals were handled according to approved institutional animal care and use committee (IACUC) protocols (#10424) of Stanford University. The protocol was approved by the Administrative Panel on Laboratory Animal Care (APLAC) of Stanford University. All surgery was performed under avertin or isoflurane anesthesia, and every effort was made to minimize suffering.

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
