## [Decision Letter]

Thank you for sending your work entitled “Phosphodiesterase 4D Acts Downstream of Neuropilin to Control Hedgehog Transduction and the Growth of Medulloblastoma” for consideration at *eLife*. Your article has been favorably evaluated by Tony Hunter (Senior Editor), a Reviewing Editor, and two reviewers.

The Reviewing Editor and the reviewers discussed their comments before reaching this decision, and the Reviewing Editor has assembled the following comments to help you prepare a revised submission.

Hedgehog (Hh) signaling pathway is a major developmental signaling pathway that governs embryonic development and adult tissue homeostasis, and its deregulation leads to birth defects and cancers including medulloblastoma (MB). Although Smo inhibitors have been approved for the treatment of Hh-related cancers, acquired drug resistance due to Smo mutations makes it necessary to develop drugs that block a downstream signaling event. A previous study by Matt Scott and his colleagues identified Neuropilin1 (Nrp1) as a positive regulator of Shh pathway but the molecular mechanism remains unknown. In this study, the authors identified the PDE4D phosphodiesterase as a downstream effector of Nrp1 in the regulation of Shh signaling. They provided evidence that Sema3F, a ligand for Nrps, stimulates the binding of PDE4D to Nrp1 and thus plasma membrane recruitment of PDE4D, leading to the degradation of cAMP at its membrane production site and consequently reduced PKA activity, which antagonizes Hh signaling, both at the centrosome and in the cytosol. Consistent with this model, pharmacological inhibitors of PDE4 inhibit Shh signaling, GNP proliferation, and medulloblastoma growth. The reviewers find the work potentially significant as it reveals a new regulatory mechanism of Hh signal transduction and implicates PDE4D as a promising target to treat Hh-related tumors. However, the reviewers also expressed substantial concerns that should be addressed by additional experiments and/or by discussion before publication is recommended.

1) A major concern is the physiological relevance of Sema3F as an Npr1 in the regulation of Shh signaling and medulloblastoma because most experiments on Sema3F are through overexpression. For example, it is not clear whether Sema3F is expressed or upregulated in medulloblastoma. The authors used the Nrp^panA^ antibody, which blocked Sema3-Nrp interaction, to demonstrate the involvement of Sema3F. However, this antibody may block the interaction of Npr1 with other ligands. The authors should employ RNAi knockdown or genetic knockout (if existing) to address whether Sema3F is the ligand for Npr1 in the regulation of Shh signaling and medulloblastoma growth.

2) A recent study (Snuderl et al., Cell 2013) suggested that placental growth factor (PLGF) acts through Npr1 to promote the growth of medulloblastoma through ERK signaling. Indeed, Snuderl et al. showed that both PLGF and Npr1 were upregulated in medulloblastoma and that blocking PLGF/Npr1 impeded tumor growth and spread. The authors should discuss their results in light of the findings by Snuderl et al. Do different classes of Ngr ligands act through distinct pathways?

3) It appears that influence of Sema3/Nrp is modulatory since Shh can still signal in its absence but if the net effects of Sema3/Nrp are severe in all or most tissues the authors could argue that the Nrp contribution is central to the efficacy of Hh signaling. Further evidence on this point and a clear statement of the authors' perspective on whether the proposed Sema-Nrp-PDE4D pathway is permissive or regulated by Shh would be valuable.

4) The reviewers are concerned about the appropriateness of cAMP and PKA measurement, especially the use of pT197 antibody. It is therefore suggested that the authors should consider measurement of Gli3 processing as a functional readout for PKA activity change under conditions of eliminating Sema3/Nrp1 contributions both with and without Shh stimulation.

cAMP: The text describes a “cell-based assay” and cites a reference concerning FRET reporters, but the assay used is of cell extracts added to purified PKA and measuring ATP usage (i.e. quite different). Figure 2—figure supplement 1 shows relative cAMP levels. A scale of absolute concentration would be far more valuable because that could be compared to the normal PKA activation profile and it also addresses the emerging picture of whether endogenous Sema3/Nrp signaling holds cAMP levels “unusually” low. I am not reassured of the efficacy of the cAMP assay by the controls presented, because I find it surprising that forskolin treatment only increases cAMP levels two-fold and similarly to constitutive Nrp activity.

PKA activity: There are difficulties measuring PKA activity in cells but the assays offered are not compelling. T197 phosphorylation is recognized as an autophosphorylation required for full activity. That could be extrapolated to imagine that pT197 might be a measure of PKA activity. However, it is generally thought (including in the very reference cited indirectly – Barzi cites Iyer et al., 2005) that T197 phosphorylation is constitutive and not regulated. Thus, for example, in many cells most catalytic subunit is complexed to regulatory subunit and inactive but is phosphorylated at T197. Even results in Figure 3 illustrate this: pPKA levels are increased only by 30% in response to forskolin. Thus, pT197 is not a universally recognized marker of PKA activity and I am not aware of convincing evidence that this view should change. The second measure of PKA activity is the use of a PKA consensus phosphopeptide antibody on Western blots. This has been used before (for example in Barzi et al.) and it likely has enough specificity for PKA substrates based on similar bands for Nrp treatments and forskolin but the 5 or so bands affected are not known (or described) and it is clearly hard to convert those results into any quantitative measure, let alone ruling out other possible contributions to the observed changes. So, I believe this measurement is a useful guide but neither definitive nor quantitative. As mentioned earlier, and in part because of the systematic difficulty of measuring PKA activity in vivo, an important supplement is to measure the net effect of PKA activity targeted towards the most relevant substrates (possibly reflecting local PKA activities) by measuring Gli processing.

[Editors' note: further revisions were requested prior to acceptance, as described below.]

Thank you for submitting your work entitled “Phosphodiesterase 4D Acts Downstream of Neuropilin to Control Hedgehog Transduction and the Growth of Medulloblastoma” for peer review at *eLife*. Your submission has been favorably evaluated by Tony Hunter (Senior Editor), a Reviewing Editor, and two reviewers.

The reviewers have discussed the reviews with one another and the Reviewing Editor has drafted this decision to help you prepare a revised submission.

The authors have taken considerable efforts to address the reviewers' comments from the initial review. There remains some reservations from one reviewer concerning whether the data sufficiently support the authors' conclusions.

Essential revisions:

Generally, the authors should strengthen their presentation and discussion to acknowledge some of the key caveats and provide alternative explanation:

1) The authors should argue why pT197 and Gli2 localization may be valid measures of PKA activity rather than making it sound like these are already accepted measures.

2) Reconciling the differential effect on Gli2 localization and Gli3 processing with Nrp1/2 knockdown. The authors invoke a PKA-independent mechanism to explain Gli3 processing but alternative possibilities should also be considered (see for example comments by Reviewer #2).

3) There remains reservation that the normal role of Nrp ligands has not been appropriately addressed by loss-of-function genetics. This caveat needs to be acknowledged, along with a previous publication from the authors that reported no effect of Sema3.

Reviewer #1:

The authors have responded to several points with some improved and additional data plus tidier descriptions. However, I find that the revised manuscript has not fully addressed some points.

The concern about physiological relevance of Sema3 stimulation of Nrps seems to have been largely misunderstood. The question is whether any Sema3 ligand normally significantly stimulates Nrps to affect Hh signaling, hence the request from all reviewers to reduce Sema3 ligands genetically. I still do not understand why this cannot usefully be done. The panA antibodies might seem to serve the same purpose but antibody binding could have additional consequences: blocking binding of a co-receptor or triggering a change in Nrp directly. I also still believe it is appropriate for the authors to state that they previously reported no effect of Sema3, with or without an explanation.

Most concerns about cAMP and PKA remain and are in fact enhanced by new data. While there is room for plenty of complexity regarding cAMP and PKA actions, it is nevertheless surprising that a large effect on the pathway supposedly mediated by changes in cAMP and PKA has no effect at all on Gli3 processing. This test was performed because direct measures of cAMP and PKA did not convince all reviewers. Those tests remain basically the same, including the unqualified assertion that pT197 is an accepted measure of PKA activation. I think the authors could use their own data with forskolin etc. to try to convince readers that pT197 indicates activity but I think it is not acceptable to state this is an accepted, justified measure. It also remains odd that Figure 3 is described as a dramatic reduction in phosphorylation when you look at the adjacent forskolin lane.

The new measures of Gli2 localization are potentially interesting, as is the point brought up by the authors of Tuson et al. and related studies that PKA significantly affects Gli activation as well as Gli processing. Concerning the data only, we see a measure of Gli2 intensity at cilia tips but not elsewhere. Is the reduction because Gli2 is at the cilium base or because there is less Gli2 total? The authors state that they have measured a transport defect. Concerning models and explanations, the discussed precedents are complex and cannot be distilled to Gli2 at the tip being a measure of PKA activity. Complete loss of PKA activity was shown to promote Gli2 accumulation at tips. Stimulation by forskolin was reported as doing the opposite but Tuson report that the effect is still seen in PKA-null animals and therefore independent of PKA. At present, I find explanations for the Gli2 and Gli3 observations rather implausible (especially a PKA-independent Gli3 processing mechanism suddenly kicking in at 100% efficiency when cAMP is altered by Nrp). If different pools of cAMP and PKA are responsible it is not obvious where they would reside. The cilium or its base appears to be a key site for regulated AC activity and PKA activity regarding Gli processing and likely for Gli localization. Does pT197 antibody show no change in cilia or is there other evidence for different spatial effects of Nrp? Altogether, I find the Nrp-PDE-cAMP-PKA-Gli connections far from resolved and I believe it is important not to represent these as proven and straightforward.

Reviewer #2:

In the revised manuscript, the authors have addressed most of my concerns adequately. One minor point is that, when discussing the differential effect of Nrp1/2 RNAi on Gli3 processing and Gli2 inhibition, the authors may want to consider a third possibility, i.e., Gli3 processing and Gli2 inhibition could be sensitive to different levels of PKA activity. For example, in Npr1/2 RNAi cells, Shh could still inhibit ciliary PKA activity via down-regulating Gpr161 to block Gli3 processing; however, the residual PKA activity might still be able to partially suppress Gli2 activity. In this view, the Sema3/Nrp1/2/PDE4 pathway acts in parallel with the Shh/SmoGpr161 pathway to achieve a more dramatic downregulation of ciliary PKA activity, leading to more robust Gli2 activation.

---

## [Author Response]

*1) A major concern is the physiological relevance of Sema3F as an Npr1 in the regulation of Shh signaling and medulloblastoma because most experiments on Sema3F are through overexpression*.

We thank the reviewers for raising this important point. In our previous study, we showed that both Nrp1 and Nrp2 regulate Hh signaling (26). In new experiments, we have now found that all class 3 semaphorin proteins (Sema3A, Sema3B, Sema3C, Sema3E, and Sema3F) can promote Hh signaling via Nrp1 or Nrp2 (or both). We have included these data in Figure 1—figure supplement 1 in the revised manuscript. The original version of the manuscript may have made it seem that only Sema3F-Nrp1 interactions are able to affect Hh transduction. In the revised manuscript, we have made clear that activation of Nrp signaling by any of multiple Sema ligands will trigger the downstream signaling which involves PDE4D membrane translocation and PKA inhibition. Given the different patterns of *Sema3* expression, our new data indicate that, in vivo, the convergence of Sema and Hh signaling could happen in different tissue contexts and at different stages of development.

We have also added new experimental evidence for the Nrp2-PDE4D interaction in Figure 2—figure supplement 1 of the revised manuscript. Therefore, a general picture has emerged of class 3 semaphorin ligands engaging either Nrp1 or Nrp2 (or both), and transducing Hh signaling via PDE4D/Nrp interactions at the plasma membrane.

In most of our experiments, except the ones in Figure 2, we stimulated cells with Sema3F because it had the strongest effect in promoting Hh transduction compared to other Sema 3 ligands we tested (Figure 1). In Figure 2, we stimulated the cells with a combination of Sema3A and 3F (we corrected the labeling in the revised manuscript).

We now show data describing the effects of all of the class 3 Semas, in the revised manuscript supplement.

*For example, it is not clear whether Sema3F is expressed or upregulated in medulloblastoma*.

Both Sema3A and Sema3F are highly expressed in Purkinje neurons in the developing cerebellum, in a pattern similar to that of *Shh* (see Figure 8). It is possible that during cerebellar development Sema3 molecules are released together with Shh to influence GNP proliferation. In addition to the increased expression level of *Nrp1* and *Nrp2* in medulloblastoma (49), the expression of multiple Sema3 ligands (Sema3A/F/C) is increased in MB based upon RNA-seq (see Figure 9) (14). Sema3A and 3F have been shown to bind to either Nrp1 or Nrp2. Their expression patterns suggest that Sema3-Nrp signaling occurs at the right time and right locations to regulate Hh signal transduction. This point has been clarified in the text

Author response image 1.Sema3 and Shh are co-expressed in Purkinje neurons in the developing cerebellum.Multiple Sema3 isoforms are co-expressed with Shh in Purkinje neurons in the developing cerebellum. In situ hybridization results are from Allen Brain Atlas. Arrows point to the cell bodies of Purkinje neurons.**DOI:**
http://dx.doi.org/10.7554/eLife.07068.021

Author response image 2.Sema3 expression levels are elevated in medulloblastoma.Multiple isoforms of Sema3 expression levels are elevated in medulloblastoma. Data are RNA-seq results from 204 patient samples, and were originally published in Cho et al., J. Clin. Oncol. 2011, 29: 1424*.***DOI:**
http://dx.doi.org/10.7554/eLife.07068.022

*The authors used the Nrp*^*panA*^
*antibody, which blocked Sema3-Nrp interaction, to demonstrate the involvement of Sema3F. However, this antibody may block the interaction of Npr1 with other ligands. The authors should employ RNAi knockdown or genetic knockout (if existing) to address whether Sema3F is the ligand for Npr1 in the regulation of Shh signaling and medulloblastoma growth*.

As detailed above, multiple Sema3 ligands may well simultaneously or in succession regulate Hh signaling. It is technically impossible to genetically remove all the Sema3 molecules at once, which is why we used the antibody. Similarly we do not expect simultaneous RNA interference with several Sema3 RNA pools is likely to give a compelling result. We agree with the reviewers that the Nrp^panA^ antibody may be blocking more than one Sema3 ligand. The antibody binds to the A1-A2 domain in Nrp1 and Nrp2, thereby blocking all Sema3 interaction with Nrps. This antibody has been extensively tested and validated (2; 36; 40). From a clinical viewpoint, this broad-spectrum blockade would be advantageous in treating MB, since multiple Sema3 ligands are highly expressed in MB (see Figure 9). Therefore, although we appreciate the reviewers’ suggestion of genetic knockout or RNAi knockdown of Sema3F, we think the suggested experiment will not convincingly impede Shh transduction due to the presence of multiple Sema3 ligands and the redundancy of Sema3/Nrp interactions, as detailed above and in the revised manuscript. We did, however, perform functional experiments using RNAi knockdown of both *Nrp1* and *Nrp2* in NIH3T3 cells (Figure 1 and Figure 3), and using genetic knockout (*Nrp2* complete knockout; *Nrp1* conditional knockout) in GNPs in the developing cerebellum (Figure 6).

2) A recent study (Snuderl et al., Cell 2013) suggested that placental growth factor (PLGF) acts through Npr1 to promote the growth of medulloblastoma through ERK signaling. Indeed, Snuderl et al. showed that both PLGF and Npr1 were upregulated in medulloblastoma and that blocking PLGF/Npr1 impeded tumor growth and spread. The authors should discuss their results in light of the findings by Snuderl et al. Do different classes of Ngr ligands act through distinct pathways?

We were excited to see their study. Both studies highlight the involvement of Nrps in the development of MB and point at Nrp1 as a promising therapeutic target. However, the two studies focused on different aspects of MB. Two important points: 1) Snuderl et al. used D283-MED and D341-MED human MB cells, which are classified as group 3 and group 4 MB, respectively. Groups 3 and 4 MBs are *not* MBs derived from damage to Hh transduction. We, on the other hand, focused on the Shh group of MB. 2) It is known that different classes of Nrp ligands activate distinct downstream signalling cascades (43). The ligand Sema3 activates the small GTPase of the Rho family, and regulates the reorganization of actin cytoskeleton in neurons. This signaling cascade is implicated in growth cone collapse (13; 45; 60). Sema3 also inhibits PKA activity in growing axons via unknown mechanisms (42; 48). On the other hand, VEGF family proteins signal through phospholipase C (PLC) to activate the prosurvival/ pro-growth factor ERK1/2 and PI3K/Akt, and contribute to endothelial stimulation and pathologic angiogenesis (7; 50). PLGF is a membrane of the VEGF protein family. Therefore Snuderl et al. and our current study report distinct signaling mechanisms downstream of Nrps, triggered by different ligands. We have revised the discussion section accordingly to emphasize the fact that different classes of Nrp ligands act through distinct pathways, and that the pathways discovered by Snuderl et al. and by us both contribute to pathogenesis of MB. We have added this information to the Discussion.

*3) It appears that influence of Sema3/Nrp is modulatory since Shh can still signal in its absence but if the net effects of Sema3/Nrp are severe in all or most tissues the authors could argue that the Nrp contribution is central to the efficacy of Hh signaling. Further evidence on this point and a clear statement of the authors' perspective on whether the proposed Sema-Nrp-PDE4D pathway is permissive or regulated by Shh would be valuable*.

The universality of Sema3-Nrp as an influence on Hh transduction is a question that will be addressed by future studies of many cell types. Based on expression of the relevant molecules, and on the observations in our 2011 paper, we expect that Sema3-Nrp influences Hh transduction in some cases but not all. It may be that Sema3-Nrp inputs have modulating effects on Hh in some cell types and permissive effects (i.e. be an essential requirement) in others. For MB and 3T3 cells, we agree with the reviewers’ interpretation that Sema3-Nrp signaling is modulatory rather than permissive for Hh transduction. Sema3-Nrp signaling does have a powerful effect on Hh transduction. The following evidence supports this idea:

1) Sema3 by itself cannot activate Hh transduction (Figure 1), but does enhance Hh signaling that has been activated by Shh or SAG (Figure 1, Figure 1—figure supplement 1).

2) During the development of many tissues, Sema3 and Shh are co-expressed, and the Hh pathway functions together with Nrps:

(A) In the developing neural tube, both Sema3 and Shh act together to control the midline crossing of the commissural axons (42).

(B) In the developing skin where the Hh signaling are critical for the development of hair follicles, Nrps are co-expressed with the Hh signalling components such as Smo in hair follicle cells (26).

(C) In the developing cerebellum Sema3 proteins are co-expressed with Shh in Purkinje neurons, and they may be released together to stimulate the proliferation of granule neurons precursors in EGL (Figure 8).

(D) Genetic knockout of both Nrps blocks Sema3 signaling and impairs GNP proliferation in vivo.

(E) Sema3 is expressed in NIH3T3 cells used in our in vitro study (Figure 1—figure supplement 1), and blocking this endogenous Sema3 with function blocking Nrp antibody (Nrp^panA^) decreased the Hh signal transduction (Figure 1).

3) In all types (genetically defined groups) of medulloblastoma, *Sema3* and *Nrp1* expression levels are significantly elevated (Figure 9). This elevated expression may contribute to increased Hh transduction that promotes tumor growth.

We have added relevant discussion to clearly state our interpretation of the results and our perspective to the revised manuscript (paragraph two in the Discussion).

*4) The reviewers are concerned about the appropriateness of cAMP and PKA measurement, especially the use of pT197 antibody. It is therefore suggested that the authors should consider measurement of Gli3 processing as a functional readout for PKA activity change under conditions of eliminating Sema3/Nrp1 contributions both with and without Shh stimulation*.

*cAMP: The text describes a “cell-based assay” and cites a reference concerning FRET reporters, but the assay used is of cell extracts added to purified PKA and measuring ATP usage (i.e. quite different).*
Figure 2—figure supplement 1
*shows relative cAMP levels. A scale of absolute concentration would be far more valuable because that could be compared to the normal PKA activation profile and it also addresses the emerging picture of whether endogenous Sema3/Nrp signaling holds cAMP levels “unusually” low. I am not reassured of the efficacy of the cAMP assay by the controls presented, because I find it surprising that forskolin treatment only increases cAMP levels two-fold and similarly to constitutive Nrp activity*.

We apologize for providing the wrong reference in the previous version of the manuscript. This assay has been used in many previous studies, and we have fixed the citation in the revised manuscript.

We agree with the reviewers that knowing the absolute concentration of cAMP in the cells could be interesting. The lack of a reliable well-established assay to measure absolute cAMP concentration in intact cells makes it impractical, while measuring absolute cAMP concentration in extracts is subject to artifacts of enzymatic influences that, during extraction, alter apparent cAMP amounts. For our study the most valuable information is the cAMP concentration change in response to perturbations in gene expression. Knowing the absolute cAMP concentration in the cell, even if feasible, would not add much to our understanding of how Sema3-Nrp regulates the Hh pathway. In general it would be revealing to know absolute concentrations of the proteins and small molecules in Hh and other pathways, in specific subcellular locations, and we look forward to the day when this is more routine. In the meantime the cAMP-Glo assay used in our study provides the change of cAMP concentration (Δ[cAMP]) after drug treatment or RNAi. In the revised manuscript we provide the cAMP standard curve generated from purified cAMP. We also provide the Δ[cAMP] in the cell lysate which was inferred from the change of luminescence intensity (ΔI = I_untreated cells_ – I_treated cells_). Please note that a negative ΔI in Sema3-treated cells indicates decrease in cAMP level compared to untreated cells. We have updated the data presentation format in the revised manuscript (Figure 2—figure supplement 2).

To address the concern of the reviewer about the efficacy of this cAMP assay in forskolin control, we optimized our experimental conditions, i.e. plating cells in the appropriate density and adding PDE4 inhibitors in cell lysis buffer to prevent cAMP from degrading. We were been able to see a more significant cAMP increase driven by forskolin after optimization. These new results are included in the revised manuscript (Figure 2—figure supplement 2).

*PKA activity: There are difficulties measuring PKA activity in cells but the assays offered are not compelling. T197 phosphorylation is recognized as an autophosphorylation required for full activity. That could be extrapolated to imagine that pT197 might be a measure of PKA activity. However, it is generally thought (including in the very reference cited indirectly – Barzi cites Iyer et al., 2005) that T197 phosphorylation is constitutive and not regulated. Thus, for example, in many cells most catalytic subunit is complexed to regulatory subunit and inactive but is phosphorylated at T197. Even results in*
Figure 3
*illustrate this: pPKA levels are increased only by 30% in response to Forskolin. Thus, pT197 is not a universally recognized marker of PKA activity and I am not aware of convincing evidence that this view should change. The second measure of PKA activity is the use of a PKA consensus phosphopeptide antibody on Western blots. This has been used before (for example in Barzi et al.) and it likely has enough specificity for PKA substrates based on similar bands for Nrp treatments and Forskolin but the 5 or so bands affected are not known (or described) and it is clearly hard to convert those results into any quantitative measure, let alone ruling out other possible contributions to the observed changes. So, I believe this measurement is a useful guide but neither definitive nor quantitative. As mentioned earlier, and in part because of the systematic difficulty of measuring PKA activity in vivo, an important supplement is to measure the net effect of PKA activity targeted towards the most relevant substrates (possibly reflecting local PKA activities) by measuring Gli processing*.

We appreciate the reviewers pointing out the difficulty of direct measurement of PKA activity. As suggested by the reviewers, we have now measured the effects of PKA on its most relevant substrates in Hh signaling – Gli2 and Gli3, and have added these new data to the revised manuscript (Figure 3—figure supplement 2).

1) PKA is known to inhibit Gli2 translocation to the cilia tip, a process required for Gli2 activation (55). To evaluate the involvement of Sema3-Nrp signaling in Gli2 translocation, we measured Gli2 intensity at cilia tips. Gli2 translocation to the cilia tips was significantly impaired after both Nrps had been silenced by lentivirus-mediated RNAi (Figure 3—figure supplement 2). Therefore, under physiological conditions, Sema3-Nrp mediated PKA inhibition promotes Gli2 translocation to the cilia tips.

2) It has been shown that PKA-driven phosphorylation of Gli3 leads to proteolytic processing of Gli3 into Gli3R, which is a transcription repressor of Hh targets. We measured the processing of Gli3 after both Nrps had been silenced by lentivirus-mediated RNAi (Figure 3—figure supplement 2). We did not see a significant change in Gli3 processing compared to cells treated with control shRNA. To clarify the different impact of Sema3-Nrp signaling on Gli2 and Gli3, we added the following discussion about the results: “This divergence in the regulation of Gli2 and Gli3 suggests that the two transcription factors may be controlled by different pools of PKA. [...] These findings suggest that Gli2 and Gli3 are independently regulated during Hh signal transduction.”

*[Editors' note*: *further revisions were requested prior to acceptance, as described below.]*

*1) The authors should argue why pT197 and Gli2 localization may be valid measures of PKA activity rather than making it sound like these are already accepted measures*.

As suggested by Reviewer 1, we have changed the corresponding text in our manuscript to reflect the point that PKA phosphorylation at T197 is not a generally accepted measure of PKA activity. We also explained why we believe the measure to be useful and meaningful in our experiments (in the subsection “Sema3-NRP signalling inhibits PKA activity). Similarly, Gli2 enrichment to the cilia tip is examined to reflect the PKA activity, but not used as a direct measurement. We have provided a detailed answer to address this question in our response to Reviewer 1 below (concerns 6, 7, 8 and corresponding answers).

*2) Reconciling the differential effect on Gli2 localization and Gli3 processing with Nrp1/2 knockdown. The authors invoke a PKA-independent mechanism to explain Gli3 processing but alternative possibilities should also be considered (see for example comments by Reviewer #2)*.

We thank the Reviewer #2 for the important suggestion. We have added this additional possibility to the revised manuscript, in the Discussion (last paragraph of subsection “The Sema3-Nrp-PDE4D pathway provides a new regulatory mechanism of PKA in Hh transduction”).

*3) There remains reservation that the normal role of Nrp ligands has not been appropriately addressed by loss-of-function genetics. This caveat needs to be acknowledged, along with a previous publication from the authors that reported no effect of Sema3*.

To do the crosses to genetically eliminate all six Sema3 ligands would take many years of work and probably would not yield viable mice. Attempting to efficiently knock down six genes is also a technical problem since it is likely to be highly inefficient and therefore inconclusive. We do agree with the reviewers that it is important to mention this caveat. We have done so in the revised manuscript in the Results section (in the subsection “Sema3 enhances Hh signal transduction”).

We have provided a detailed answer to the differences in Sema3 effects observed in the previous and current studies in our answer to Reviewer #1 (concern 3). We have also included this explanation in the revised manuscript..

Reviewer #1:

*The concern about physiological relevance of Sema3 stimulation of Nrps seems to have been largely misunderstood. The question is whether any Sema3 ligand normally significantly stimulates Nrps to affect Hh signaling, hence the request from all reviewers to reduce Sema3 ligands genetically. I still do not understand why this cannot usefully be done*.

In our in vitro data, treatment with any one of the individual Sema3 proteins (e.g., A, B, C, E, F; Sema3D was not tested because the recombinant protein is not available) is sufficient to promote Hh transduction. Based on our RNA-seq data (not included in the manuscript), in many cell types and tissue (such as NIH3T3 cells and developing cerebellum), multiple isoforms of Sema3 are co-expressed. Thus, we would have to block all Sema3 isoforms in order to compromise Hh signaling. To genetically knockout all 6 isoforms would take years of work and mice heterozygous for so many mutations might well be inviable. Even if we were to make the knockout mice, it would not add much to our understanding of how Sema3-Nrp regulates Hh pathway. We do agree with the reviewer that it is important to mention this caveat. We have done so in the revised manuscript (Results, subsection “Sema3 enhances Hh signal transduction”).

*The panA antibodies might seem to serve the same purpose but antibody binding could have additional consequences: blocking binding of a co-receptor or triggering a change in Nrp directly*.

We share the same concerns of the reviewer about possible non-specific effects of antibodies. However, the Nrp^PanA^ antibody has been very well characterized. It specifically blocks Sema3 function by abolishing its interaction with Nrps (2; 36; 40), and has been used to block Nrp functions in tumors with elevated Nrp expression, thereby inhibiting tumor growth in mouse medulloblastoma model (49). Even though there could be non‐specific effects, it is unlikely that these side effects would happen to affect the Hh pathway.

*I also still believe it is appropriate for the authors to state that they previously reported no effect of Sema3, with or without an explanation*.

The reviewer is correct that a previous publication from the Scott lab reported no increase in Hh signaling when a Sema3A-Fc recombinant protein was added to Shh- LIGHT2 cells (NIH3T3 cells stably transfected with Luciferase under control of Gli regulatory elements) stimulated with Shh (Figure S9 in Hillman et al., 2012). In that study, a luciferase-based reporter of Gli activity was used as the readout of Hh signaling. In our current studies, using more sensitive qPCR measurements of *Gli* transcription in different cells (unmodified NIH3T3 cells), we see an enhancement of Hh transduction (in a Neuropilin-dependent manner) with all five of the Sema3 proteins tested. It is possible that the different batch of Sema3 recombinant protein used in the previous study was not as active as our current protein, or there may be an idiosyncrasy of that specific cell line. In any case, we see consistent effects of multiple Sema3 ligands on enhancing Hh signaling in the cell line we use now and, importantly, in primary cultured GNPs (Figure 5). We now mention the finding from the previous publication.

*Most concerns about cAMP and PKA remain and are in fact enhanced by new data. While there is room for plenty of complexity regarding cAMP and PKA actions, it is nevertheless surprising that a large effect on the pathway supposedly mediated by changes in cAMP and PKA has no effect at all on Gli3 processing. This test was performed because direct measures of cAMP and PKA did not convince all reviewers. Those tests remain basically the same, including the unqualified assertion that pT197 is an accepted measure of PKA activation. I think the authors could use their own data with forskolin etc. to try to convince readers that pT197 indicates activity but I think it is not acceptable to state this is an accepted, justified measure*.

We thank the reviewer for pointing this out. We have changed the manuscript to reflect the fact that pT197 is not a generally accepted measurement of PKA activity.

Phosphorylation of the PKA catalytic subunit at T197 has been shown to be essential for the kinase activity, as alanine mutant at this site lowers the catalytic activity by hundreds of fold (1; 51). Meanwhile, it has been asserted that phosphorylation of PKA at T197 is stable, and that PKA activity is mainly regulated by the levels of cAMP (Taylor et al., 2013). However, we and other researchers have detected increased PKA phosphorylation following forskolin-induced increases in intracellular cAMP levels (Figure 3) or after adding the cAMP analog DBA to the cells (5). These results connect intracellular cAMP levels with the levels of T197 phosphorylation. Although phosphorylation of T197 is not a direct measurement of PKA activity, its level is correlated with the cAMP level, and PKA activity, at least in the experimental conditions employed in our study. We have included this explanation in our revised manuscript to justify the use of pPKA-T197 antibody as a PKA activity detection method (in the subsection “Sema3-Nrp signalling inhibits PKA activity”).

We would like to point out that in addition to the detection of pT197-PKA level, we use other methods as indicators of PKA activity, such as PKA substrate phosphorylation (Figure 3) and Gli2 enrichment at the cilia tips (Figure 3—figure supplement 2). The results from all these assays are consistent with the results from pT197-PKA assays, and they all support the conclusion that a Sema3-mediated decrease in cAMP level inhibits PKA activity. In summary, we have used multiple independent methods showing that Sema3-mediated decrease of cAMP level correlates with a decrease in PKA activity.

*It also remains odd that*
Figure 3
*is described as a dramatic reduction in phosphorylation when you look at the adjacent forskolin lane*.

We quantified the overall intensity of all bands in 3G, and found that Sema3 decreases PKA substrate phosphorylation by ∼27%. To avoid vague adjectives, we have changed the description in the manuscript to reflect this.

The new measures of Gli2 localization are potentially interesting, as is the point brought up by the authors of Tuson et al. and related studies that PKA significantly affects Gli activation as well as Gli processing. Concerning the data only, we see a measure of Gli2 intensity at cilia tips but not elsewhere. Is the reduction because Gli2 is at the cilium base or because there is less Gli2 total?

We chose to measure Gli2 at the cilia tips because translocation of Gli2 to the cilia tips was found to be essential for Gli2 activation (29; 54). We thank the reviewer for the suggestion of checking the Gli2 levels in other locations of the cell (such as cilium base) and total Gli2 levels. The Gli2 levels at the cilium base did not show any obvious changes after Nrp knockdown (Figure 3—figure supplement 2).

It is known that the total level of Gli2 may not directly correlate with its enrichment at the cilia tips. For example in PKA-null mice, although the total Gli2 level is reduced (Tuson et al., Figure 4), Gli2 accumulation at cilia tips is heightened (Tuson et al., Figure 7). Therefore we think measuring Gli2 total level after Nrp RNAi will not add much to our understanding of how Nrps regulate Hh signal transduction. Nevertheless, we do agree that the phenotype of reduced Gli2 enrichment at cilia tips after Nrp RNAi could have causes other than a transportation defect, so we have changed “Gli2 transportation” into “Gli2 enrichment” in our revised manuscript (please see the subsection “Regulation of Gli2 and Gli3 by Sema3-Nrp-mediated inhibition of PKA activity” and the title of Figure 3—figure supplement 2).

Our major point is the link between Sema3-Nrp signaling and PDE4D. PKA activity is examined because it is downstream of PDE4D and is a well-established Hh transduction regulator. We feel that it is beyond the scope of our current study to investigate why the total Gli2 level does not correlate with Gli2 enrichment at cilia tips.

*The authors state that they have measured a transport defect. Concerning models and explanations, the discussed precedents are complex and cannot be distilled to Gli2 at the tip being a measure of PKA activity. Complete loss of PKA activity was shown to promote Gli2 accumulation at tips. Stimulation by forskolin was reported as doing the opposite but Tuson report that the effect is still seen in PKA-null animals and therefore independent of PKA*.

Thanks to the reviewer for raising this important point. Tuson et al. found that Gli2 is enriched in PKA-null cells, suggesting that one role of PKA is in Gli2 translocation to the cilia tip ([55], Figure 7). Second, they found that adding Forskolin to PKAnull cells reduced Gli2 enrichment at the cilia tips ([55], supplementary Figure 7). As Tuson et al. suggested in their discussion, the Forskolin effect may be mediated by cAMP effectors other than PKA, such as Epac and cAMP-gated ion channels. In such an artificial condition, in which cAMP levels rise to many fold higher than physiological levels, these cAMP effectors may change cilia physiology in ways that are difficult to know, thereby indirectly changing Gli2 localization. Our data show that the Sema3-induced cAMP change is much less than the effect of Forskolin (50% in magnitude) (Figure 2—figure supplement 2), and the effect of Sema3-Nrp on Gli2 enrichment at cilia tips is presumably mainly mediated by PKA. Nevertheless, Epac and cAMP-gated ion channels may indeed have some affect upon Gli2 enrichment at cilia tips under physiological conditions. This is an interesting question for further exploration but is not the focus of our current study. We included this discussion in the subsection “Sema3-Nrp signal controls PDE4D localization to modulate Hh signal transduction”, and explained it in the first paragraph of the subsection “Sema3-Nrp signaling inhibits PKA activity”.

We agree that Gli2 accumulation at cilia tips is not a direct measurement of PKA activity. We have revised our manuscript to explain that Gli2 enrichment at the cilia tips is examined to reflect PKA activity (subsection “Regulation of Gli2 and Gli3 by Sema3-Nrp-mediated inhibition of PKA activity”). We also agree that the decrease in Gli2 accumulation at the cilia tips after Nrp RNAi may not be merely a transport defect, and we have changed the “Gli2 transportation” into “Gli2 enrichment” at cilia tips in our revised manuscript (please see the aforementioned subsection and the title of Figure 3—figure supplement 2).

*At present, I find explanations for the Gli2 and Gli3 observations rather implausible (especially a PKA-independent Gli3 processing mechanism suddenly kicking in at 100% efficiency when cAMP is altered by Nrp). If different pools of cAMP and PKA are responsible it is not obvious where they would reside. The cilium or its base appears to be a key site for regulated AC activity and PKA activity regarding Gli processing and likely for Gli localization*. *Does pT197 antibody show no change in cilia or is there other evidence for different spatial effects of Nrp?*

Although immunofluorescence data show that PKA concentrates at the cilia base, it is also distributed through the entire cytoplasm. It is not clear which subcellular PKA pool is most important for phosphorylation of Gli2/3. Although most studies have focused on PKA activity at the cilium base, we cannot exclude the possibility of PKA actions in other subcellular sites, such as in cilia, at the cytoplasmic membrane surrounding the cilia, during transportation of Gli2/3 into/out of cilia/nucleus, or other locations. Therefore it is impossible for us to pin down all the specific subcellular sites where Nrps are or are not influencing PKA activity in the Hh pathway. We have observed Nrp protein diffusely localized to the entire cytoplasmic membrane ([26], Supplemental figure S10), in keeping with past studies (Chen et al., 1997; He and Tessier-Lavigne, 1997). Nrps might localize to the cilia membrane and its localization may be too dynamic and transient to detect; but in any case we do not see it enriched there.

It is well documented that Gli2 activation and Gli3 processing is differentially regulated by PKA ([Bibr bib11a]; [Bibr bib16a]; [41]; [55]). PKA phosphorylation causes a significant fraction of Gli3 to be proteolytically processed, while only a minor fraction of Gli2 is proteolytically processed ([41]; Pan et al., 2009). PKA may affect Gli2 mainly by controlling its spatial localization relative to the cilia, and/or its activation (54; 55). Although Gli3 processing is known to be controlled by PKA, a significant fraction of Gli3 is processed into Gli3R in PKA-null mice ([55], Figure 4). Hence PKA-independent Gli3 processing exists. Here in our paper we focus on how Sema-Nrp signaling alters Hh transduction, but indeed more needs to be done to understand how PKA affects Hh transduction, especially how PKA differentially regulates Gli2 activation and Gli3 processing.

An alternative explanation (we appreciate the suggestion by Reviewer 2) is that Gli2 inhibition and Gli3 processing could be sensitive to different levels of PKA activity. This model could relate to another PKA regulator, GPR161, which resides in the cilium membrane and inhibits PKA by lowering cAMP levels at the cilium base ([Bibr bib38a]). In cells without Nrps, Shh could still be able to lower PKA activity at the cilium base by inhibiting GPR161 to block Gli3 processing, whereas residual PKA activity maybe sufficient to suppress Gli2 activation. In this model, the Sema3-Nrp-PDE4D pathway we describe here might act in parallel to Shh-GPR161 pathway to more robustly reduce PKA activity in cilium, leading to more robust Gli2 activation. In this way, the Sema3-Nrp-PDE4 pathway acts in parallel with the Shh-GPR161 pathway to achieve a more dramatic reduction of ciliary PKA activity and consequent robust Gli2 activation. We have included this explanation in the subsection “Sema3-Nrp signaling inhibits PKA activity”).

Regarding the level of pT197-PKA in the cilium, we and other groups have tried to detect pT197-PKA signals in cilia using immunofluorescence but the results have been negative (5). This could be due to very low PKA levels in cilia that it is under detection threshold, or because PKA localization in the cilium is too dynamic and transient to be detected.

*Altogether, I find the Nrp-PDE-cAMP-PKA-Gli connections far from resolved and I believe it is important not to represent these as proven and straightforward*.

We agree that the pathway is more complicated than a simple linear model of Nrp-PDE4-cAMP-PKA-Gli. The major discovery of our current study is the link between Sema3-Nrp signaling and PDE4D activity which eventually affects the intracellular cAMP level. Multiple proteins are activated by cAMP in the cell, including PKA, Epac and cAMP-gated ion channels. PKA activity is affected, and other cAMP effectors downstream of the Sema3-Nrp-PDE4D pathway may be involved. The roles of these cAMP effectors in Hh signal transduction remain to be explored. We mentioned this caveat in our original Discussion, and we have added further explanation to reflect the reviewer’s point in the revised manuscript (in the subsection “Sema3-Nrp signaling inhibits PKA activity”).

Reviewer #2:

*In the revised manuscript, the authors have addressed most of my concerns adequately. One minor point is that, when discussing the differential effect of Nrp1/2 RNAi on Gli3 processing and Gli2 inhibition, the authors may want to consider a third possibility, i.e., Gli3 processing and Gli2 inhibition could be sensitive to different levels of PKA activity. For example, in Npr1/2 RNAi cells, Shh could still inhibit ciliary PKA activity via down-regulating Gpr161 to block Gli3 processing; however, the residual PKA activity might still be able to partially suppress Gli2 activity. In this view, the Sema3/Nrp1/2/PDE4 pathway acts in parallel with the Shh/SmoGpr161 pathway to achieve a more dramatic downregulation of ciliary PKA activity, leading to more robust Gli2 activation*.

Thanks for this important suggestion. We have incorporated this model into our revised manuscript (subsection “The Sema3-Nrp-PDE4D pathway provides a new regulatory mechanism of PKA in Hh transduction”).